

# A new juvenile *Yamaceratops* (Dinosauria, Ceratopsia) from the Javkhlant Formation (Upper Cretaceous) of Mongolia

Minyoung Son[1,2], Yuong-Nam Lee[1], Badamkhatan Zorigt[3], Yoshitsugu Kobayashi[4], Jin-Young Park[1], Sungjin Lee[1], Su-Hwan Kim[1] and Kang Young Lee[5]

[1] School of Earth and Environmental Sciences, Seoul National University, Seoul, South Korea
[2] Department of Earth and Environmental Sciences, University of Minnesota, Minneapolis, Minnesota, United States
[3] Institute of Paleontology, Mongolian Academy of Sciences, Ulaanbaatar, Mongolia
[4] Hokkaido University Museum, Hokkaido University, Sapporo, Japan
[5] Department of Physics Education, Gyeongsang National University, Jinju, South Korea

## ABSTRACT

Here we report a new articulated skeleton of *Yamaceratops dorngobiensis* (MPC-D 100/553) from the Khugenetjavkhlant locality at the Shine Us Khudag (Javkhlant Formation, ?Santonian-Campanian) of the eastern Gobi Desert, Mongolia, which represents the first substantially complete skeleton and the first juvenile individual of this taxon. The specimen includes a nearly complete cranium and large portions of the vertebral column and appendicular skeleton. Its skull is about 2/3 the size of the holotype specimen, based on mandibular length. Its juvenile ontogenetic stage is confirmed by multiple indicators of skeletal and morphological immaturity known in ceratopsians, such as the long-grained surface texture on the long bones, the smooth external surface on the postorbital, open neurocentral sutures of all caudal vertebrae, a large orbit relative to the postorbital and jugal, the low angle of the lacrimal ventral ramus relative to the maxillary teeth row, narrow frontal, and straight ventral edge of the dentary. Osteohistological analysis of MPC-D 100/553 recovered three lines of arrested growth, implying around 3 years of age when it died, and verified this specimen's immature ontogenetic stage. The specimen adds a new autapomorphy of *Yamaceratops*, the anteroventral margin of the fungiform dorsal end of the lacrimal being excluded from the antorbital fossa. Furthermore, it shows a unique combination of diagnostic features of some other basal neoceratopsians: the ventrally hooked rostral bone as in *Aquilops americanus* and very tall middle caudal neural spines about or more than four times as high as the centrum as in *Koreaceratops hwaseongensis*, *Montanoceratops cerorhynchus*, and *Protoceratops andrewsi*. The jugal with the subtemporal ramus deeper than the suborbital ramus as in the holotype specimen is also shared with *A. americanus*, *Liaoceratops yanzigouensis*, and juvenile *P. andrewsi*. Adding 38 new scorings into the recent comprehensive data matrix of basal Neoceratopsia and taking into account the ontogenetically variable characters recovered *Y. dorngobiensis* as the sister taxon to Euceratopsia (Leptoceratopsidae plus Coronosauria). A second phylogenetic analysis with another matrix for Ceratopsia also supported this position. The new phylogenetic position of *Y. dorngobiensis* is important in ceratopsian evolution, as

Corresponding author
Yuong-Nam Lee, ynlee@snu.ac.kr

this taxon represents one of the basalmost neoceratopsians with a broad, thin frill and hyper-elongated middle caudal neural spines while still being bipedal.

## INTRODUCTION

Ceratopsian dinosaurs appeared in the Late Jurassic of Asia and flourished in the Late Cretaceous of North America. In early evolutionary history, basal ceratopsians were small, bipedal dinosaurs without much-elaborated structures (*Xu et al., 2006*; *Han et al., 2016*), but the Late Cretaceous ceratopsoids became quadrupedal giants with large horns and frills on their heads (*Sues & Averianov, 2009*; *Wolfe et al., 2010*). Intermediate between the non-neoceratopsian ceratopsians and ceratopsoids are the non-ceratopsoid neoceratopsians, referred to as basal neoceratopsians (*You & Dodson, 2004*).

The earliest fossil record of ceratopsian dinosaurs appears to be two Jurassic taxa, *Yinlong* and *Hualianceratops*, both from the Upper Jurassic (Oxfordian) Shishugou Formation (*Xu et al., 2006*; *Han et al., 2015*, *2016*). The transition from basalmost ceratopsians (represented by *Yinlong*) to ceratopsids (represented by *Triceratops*) has encompassed many anatomical innovations through the transformational and step-wise acquisition of unique traits. These include bony structures such as the enlarged frill and horns related to display (*Prieto-Márquez et al., 2020*), complex dental battery for food processing (*Erickson et al., 2015*), and even neomorphic ossifications for keratinous coverings such as the rostral, epijugal, epinasal, episquamosal, and epiparietal (*Horner & Goodwin, 2008*). Ceratopsians are also interesting in that during their evolutionary history, they show an increase in body size, bipedal to quadrupedal transition, and dispersal from Asia to Laramidia, Europe, and Appalachia.

Among basal neoceratopsians is *Yamaceratops dorngobiensis* from the Upper Cretaceous Javkhlant Formation of eastern Mongolia (*Makovicky & Norell, 2006*; *Eberth et al., 2009*). Although represented by a holotype partial skull and referred disarticulated elements, the exact osteology of *Yamaceratops* is not known yet due to the absence of articulated postcranial skeletons. Thus, the phylogenetic relationships of *Yamaceratops* among basal neoceratopsians have not been entirely solved since its initial description in 2006 (*Makovicky & Norell, 2006*; *Chinnery & Horner, 2007*; *Makovicky, 2010*; *Lee, Ryan & Kobayashi, 2011*; *Ryan et al., 2012*; *Farke et al., 2014*; *He et al., 2015*; *Zheng, Jin & Xu, 2015*; *Han et al., 2018*; *Knapp et al., 2018*; *Morschhauser et al., 2018c*; *Arbour & Evans, 2019*; *Yu et al., 2020*).

*Yamaceratops* is the only known ceratopsian from the Javkhlant Formation at the Dornogovi Province and its correlative strata at the Zos Canyon locality (*Norell & Barta, 2016*) (Fig. 1A). Other ceratopsians from the Dornogovi Province of eastern Mongolia include the "psittacosaurs" from the Dzun Shakhai locality (*Watabe et al., 2010*) and an indeterminate leptoceratopsid (PIN 4046/11; formerly "*Udanoceratops*") from the Baga
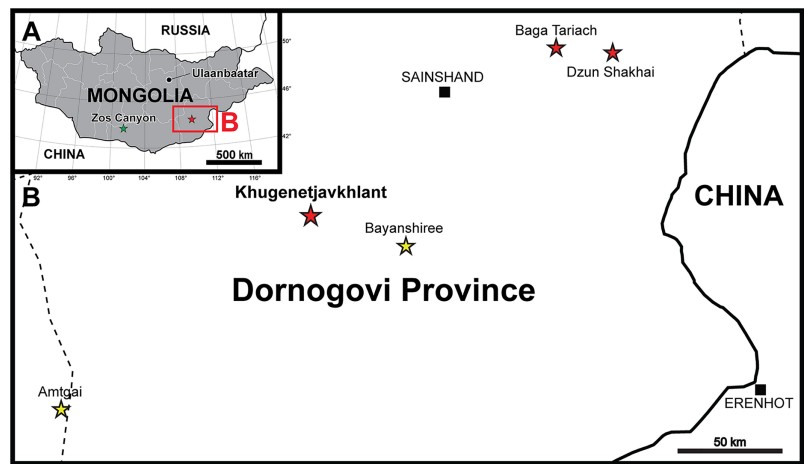

**Figure 1 Maps of the locality where the new skeleton of *Yamaceratops dorngobiensis* (MPC-D 100/553) was discovered.** (A) Map of Mongolia. Stars represent the outcrops of the Javkhlant Formation and its correlative strata; green star indicates the Zos Canyon locality and red star the Khugenetjavkhlant locality; (B) Map of Dornogovi Province. Red stars indicate ceratopsian localities. Yellow stars represent the main localities of the Bayanshiree Formation. Dashed lines represent province boundaries.                               

Tariach locality (*Tereshchenko, 2001*, *2020*) (Fig. 1B). A *Protoceratops* sp. specimen (MPC-D 100/517) from the Shurg Uul locality is likely from the "Shurguul, Sevrei sum, Omnogovi province," instead of Dornogovi Province (*Tsogtbaatar et al., 2019*: 90; contra *Czepiński, 2020*). Some protoceratopsid fossils were reported from the Baga Tariach locality (*Watabe et al., 2010*), but the specimens had not been described yet.

Ontogeny and variation in a few basal neoceratopsian taxa have been extensively studied since early research history. It was made possible by discovering many well-preserved specimens from Mongolia, as described by *Brown & Schlaikjer (1940)* for *Protoceratops andrewsi* and by *Maryańska & Osmólska (1975)* for *Bagaceratops rozhdestvenskyi*. Despite these studies, many other names have been proposed for specimens of these taxa of different growth stages, especially for juvenile specimens, which were only recently reviewed and synonymized (*Makovicky & Norell, 2006*; *Czepiński, 2019*, *2020*).

Here we describe a new articulated skeleton of a small individual of *Yamaceratops dorngobiensis* (MPC-D 100/553) discovered from the middle Javkhlant Formation (?Santonian-Campanian) at Khugenetjavkhlant locality in 2014 (Figs. 2 and 3). Khugenetjavkhlant is equivalent to the misspelled 'Khugenslavkhant' (*Eberth et al., 2009*; *Tanaka et al., 2019*) and 'Khugenetslavkant' (*Makovicky & Norell, 2006*; *Balanoff et al., 2008*; *Nesbitt et al., 2011*; *Makovicky et al., 2011*; *Varricchio, Balanoff & Norell, 2015*). It is the same locality where the holotype of *Yamaceratops* was found. We propose new diagnostic characters for this taxon and describe its articulated postcranial skeleton for the first time. We also examine the chronological age of death of this specimen through histological analysis and review the indicators of skeletal and morphological maturity (*Hone et al., 2014*; *Hone, Farke & Wedel, 2016*; *Griffin et al., 2021*). By comparing MPC-D 100/553 to the holotype and referred materials, the ontogenetically variable features of

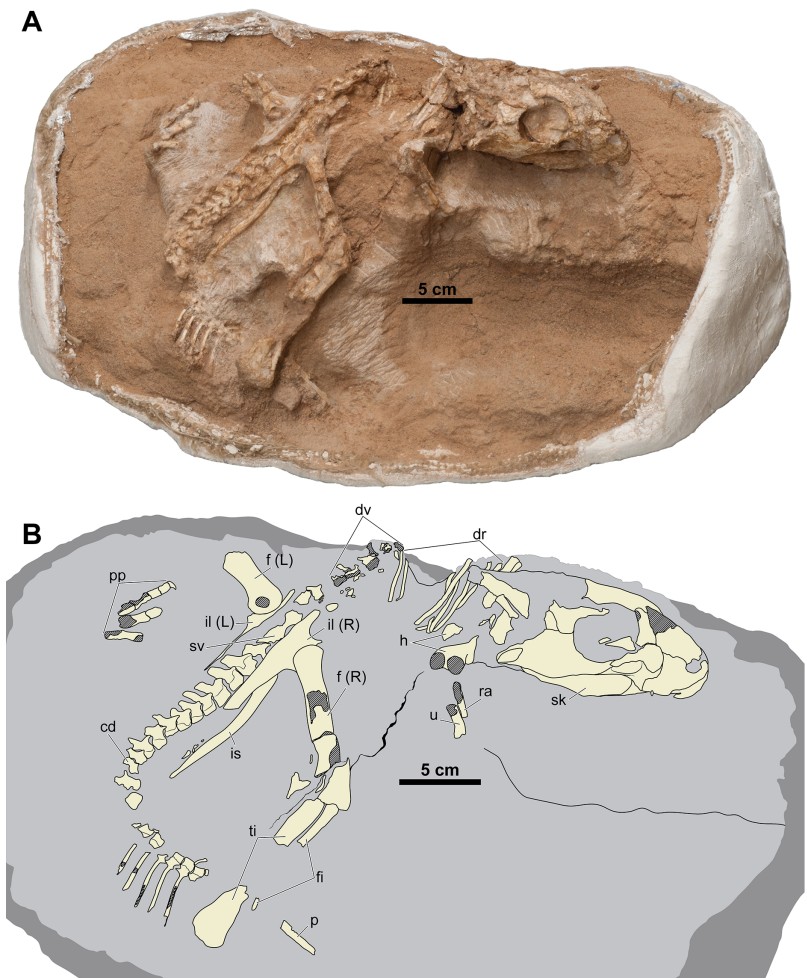

**Figure 2 Skeleton of *Yamaceratops dorngobiensis* (MPC-D 100/553) in right dorsolateral view.** (A) Photograph; (B) Interpretive drawing. Bones are bounded by solid lines and colored beige; the matrix is gray. Shaded areas represent the broken surface of bones. Abbreviations: cd, caudal vertebrae; dr, dorsal ribs; dv, dorsal vertebrae; f, femur; fi, fibula; h, humerus; L, bone on the left side; p, isolated parietal; pp, pedal phalanges; R, bone on the right side; ra, radius; sk, skull; sv, sacral vertebrae; ti, tibia; u, ulna; il, ilium; is, ischium.

*Yamaceratops* are investigated and compared with those in other well-sampled taxa. The well-preserved postcranial anatomy of MPC-D 100/553 offers insight into basal ceratopsian locomotion and evolution as well. We believe that the new osteological characters and inferred patterns of ontogenetic variation in this study can help resolve the phylogenetic position of *Yamaceratops* and clarify the character evolution of basal neoceratopsians.

## MATERIALS AND METHODS

### Phylogenetic analysis

A strict consensus tree was constructed based on the most recent iteration of the comprehensive character matrix of *Morschhauser et al. (2018c)*. The *Morschhauser et al. (2018c)* matrix had 41 taxa with all 257 characters ordered. Later, in the iteration of this

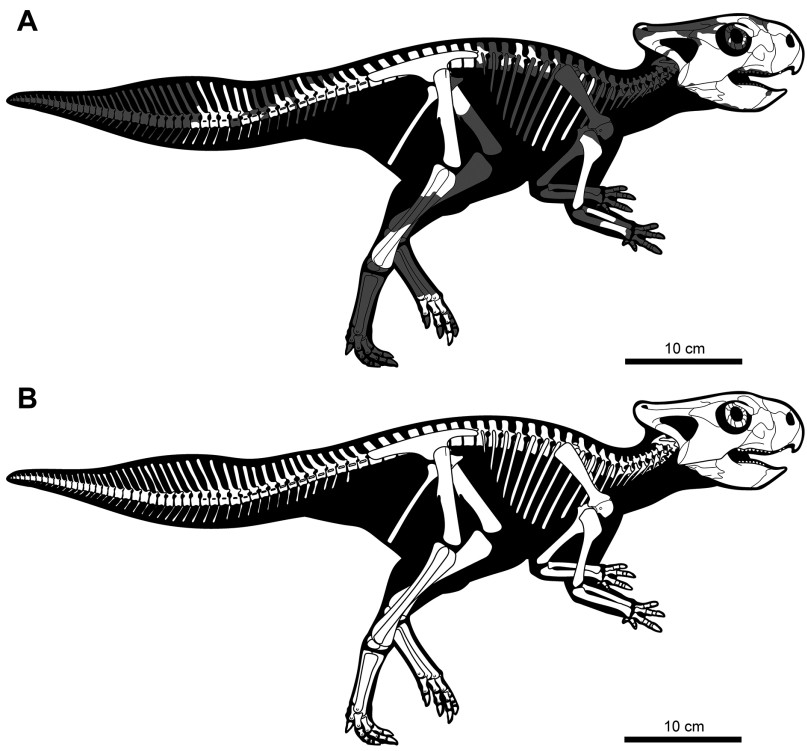

**Figure 3  Skeletal reconstruction of juvenile *Yamaceratops dorngobiensis* (MPC-D 100/553) in right lateral view.** (A) Reconstruction of the skeleton with preserved parts colored in white and missing bones in gray; (B) reconstruction of the complete skeleton, restored based on other basal neoceratopsian skeletons.                               

matrix by *Arbour & Evans (2019)*, one additional taxon (*Ferrisaurus*) was added, and all characters were treated as unordered, as we followed in this paper. We compared the ontogenetic variation within *Yamaceratops*, based on comparisons of the juvenile specimen (MPC-D 100/553) to the adult one (IGM 100/1315), with that of *P. andrewsi* (*Brown & Schlaikjer, 1940*) and found generalized patterns. Therefore, we applied them to our analysis, assuming they were widespread in basal neoceratopsians. From these observations, we found additional ontogenetic characters in the *Morschhauser et al. (2018c)* matrix, totaling nine out of 257 characters (Supplemental Information). Juvenile states in these ontogenetically variable characters were scored as '?' in taxa represented solely by juvenile specimens (*Asiaceratops*, *Aquilops*, and 'Graciliceratops') and *Yamaceratops* (MPC-D 100/553). A total of 38 characters were re-scored for *Yamaceratops* (Supplemental Information). The matrix was analyzed in TNT version 1.5 as parsimony analysis using the Traditional Search (*Goloboff & Catalano, 2016*). The search followed the method of *Morschhauser et al. (2018c)* that was iterated in *Arbour & Evans (2019)*, and the parameters are as follows: MaxTrees of 10,000 trees; *Lesothosaurus* as an outgroup taxon; tree bisection reconnection (TBR) swapping algorithm with 1,000 replications. The 'Bremer.run' script in TNT was used to calculate the Bremer support values (*Goloboff & Catalano, 2016*). The strict consensus tree was time-scaled using the R package "strap" (*Bell & Lloyd, 2015*).

## CT scanning

We made micro-CT scans by using Xradia 620 Versa (Carl Zeiss, Thornwood, NY, USA) for the right jugal (80 kV, 127 uA, pixel size 56.5881 um) and right distal humerus (80 kV, 126 uA, pixel size 46.6794 um), and by using Xradia 520 Versa (Carl Zeiss, Thornwood, NY, USA) for the articulated skull (120 kV, 84 uA, pixel size 95.2099 um). The images were segmented with Dragonfly (Object Research Systems (ORS) Inc., Montreal, Canada). The scanned images were used to clarify some aspects of morphology that could not be discerned from the surface of the specimen. Detailed analysis of the internal structure is outside the scope of this paper and will not be discussed here.

## Histological analysis

For histologic analysis, we chose the associated humerus. Before sectioning, photography, photogrammetry, and CT scanning of the humerus were conducted. Molding and casting were made from the specimen for later research. The mid-shaft of the humerus was sampled for the examination. The sample was prepared and transversely thin sectioned following *Lamm (2013)*. The slides were examined using Nikon Eclipse E600 POL petrographic polarizing microscope with a lambda 530 nm plate. The bone cross-section was photographed using a combination of Nikon DS-Ri2 camera and NIS-Elements BR (ver. 4.13) software. Adobe Photoshop (ver. 21.2) is used for image enhancement and tracing LAGs. Bone wall thickness and area of the transverse section were quantified by Image J (ver. 1.53; *Schneider, Rasband & Eliceiri, 2012*).

## RESULTS

**Systematic paleontology**

Dinosauria *Owen, 1842*

Ornithischia *Seeley, 1887*

Marginocephalia *Sereno, 1986*

Ceratopsia *Marsh, 1890*

Neoceratopsia *Sereno, 1986*

*Yamaceratops dorngobiensis Makovicky & Norell, 2006*

**Holotype.** IGM 100/1315, an articulated skull from an adult that lacks the rostral, premaxillae, nasals, the predentary, anterior process of the lacrimal, left elements around the temporal region, left posterior mandible, and the parietal (*Makovicky & Norell, 2006*).

**Studied specimen.** MPC-D 100/553, a reasonably complete articulated skeleton with the left proximal femur and partial foot, but no cervical vertebrae, pectoral girdles, left arm, right hand, other parts of the left leg, and the right foot. Measurements are in the Supplemental Information (Tables S1–S4).

**Locality, horizon, and age.** Khugenetjavkhlant, Dornogovi Province, Mongolia; middle unit of the Javkhlant Formation; Upper Cretaceous (?Santonian-Campanian). The specimen was preserved on top of a trough cross-bedding sandstone that is poorly

sorted, as was described for the typical coarse-grained deposits in the middle Javkhlant Formation (*Eberth et al., 2009*).

**Emended diagnosis.** A neoceratopsian possessing the following autapomorphies: unkeeled rostral bone ventrally hooked and posteriorly expanded; the anteroventral margin of the fungiform anterior process of the lacrimal excluded from the antorbital fossa; jugal with its subtemporal ramus deeper than the suborbital ramus with an obtuse angle in between at the ventral edge. In addition, *Yamaceratops dorngobiensis* shares the following characters that are otherwise unique among basal neoceratopsians: a ventrally hooked rostral bone as in *Aquilops americanus*; a fungiform anterior process on the lacrimal as in *Auroraceratops rugosus*; a jugal with a deeper subtemporal ramus than the suborbital ramus as in *Aquilops americanus*, *Liaoceratops yanzigouensis*, and juvenile *Protoceratops andrewsi*; middle caudal neural spines about or more than four times as high as the centrum as in *Koreaceratops hwaseongensis*, *Montanoceratops cerorhynchus*, and *Protoceratops andrewsi*.

**Remarks.** MPC-D 100/553 is identified as *Yamaceratops* based on the unique characters shared with other *Yamaceratops* specimens (IGM 100/1303 and 100/1315), such as the unkeeled and posteriorly expanded rostral bone, jugal with its subtemporal ramus deeper than the suborbital ramus with an obtuse angle in between at the ventral edge, and two tubercles on the ventral margin of the angular (*Makovicky & Norell, 2006*).

The lacrimal with a "fungiform expansion of the dorsal end" and a "concave rostroventral margin due to limited participation in the floor of the antorbital fossa" was suggested for *Auroraceratops*, *Bagaceratops*, and *Liaoceratops* (*Morschhauser et al., 2018c*; character 49). However, the expansion of the anterior ramus is not extensive in *Liaoceratops* (*You, Tanoue & Dodson, 2007*) and the ventral ramus participates in the medial wall of the antorbital fossa in *Bagaceratops* (*Czepiński, 2019*). The anterior ramus of the lacrimal is rectangular in *Beg tse* (*Yu et al., 2020*). Therefore, the fungiform expansion of the anterior ramus of the lacrimal is only distinct in *Auroraceratops* (*You et al., 2005*) and *Yamaceratops* (MPC-D 100/553). The lacrimal of MPC-D 100/553 differs from *Auroraceratops* and all other basal neoceratopsians. The anteroventral margin of the anterior ramus of the lacrimal neither participates in the floor nor the margin of the antorbital fossa. For these reasons, this character is considered autapomorphic for *Yamaceratops* herein.

The presence and number of angular tubercles are plesiomorphic for basal neoceratopsians, although previously thought to be diagnostic for *Liaoceratops* (*Xu et al., 2002*) and *Yamaceratops* (*Makovicky & Norell, 2006*). The numbers and position of the tubercles vary among taxa. Three tubercles are present in *Liaoceratops* (*Xu et al., 2002*), and two tubercles in *Mosaiceratops* are positioned more dorsally than in *Yamaceratops* (*Zheng, Jin & Xu, 2015*). However, these are variable in *Auroraceratops*, with the holotype specimen bearing two tubercles on the right angular and none on the left (*Morschhauser et al., 2018a*). Considering that the tubercles are positioned in the area where the attachment of *M. pterygoideus ventralis* is implied (*Nabavizadeh, 2020*), these tubercles

may result from varying degrees of jaw muscle development. In mammals, similar roughened tubercles are often muscle attachment sites (*e.g.*, *Clifford & Witmer, 2004*). It is also worth noting that the surangular lateral ridge was suggested as an attachment site of jaw adductor muscle for basal ceratopsians, including *Psittacosaurus* and *Protoceratops* (*Haas, 1955*; *Nabavizadeh, 2020*). In *Psittacosaurus* species, the well-developed dentary flange in adults has also been suggested as a site for adductor muscle attachment (*Sereno, Xijin & Lin, 2010*), but this idea was later criticized (*Taylor et al., 2017*).

Although an embayment at the posterior base of the dorsal process of the jugal is present on both the holotype (IGM 100/1315) and the new specimen of *Yamaceratops* (MPC-D 100/553), this is likely due to postmortem dorsoventral crushing of the skull resulting in the breakage and displacement of the thin dorsal process of the jugal, judging from the micro-CT images of the jugal in MPC-D 100/553. Another possible example of such displacement of the jugal dorsal process due to crushing can be seen in *Aquilops* (*Farke et al., 2014*).

**Ontogenetic Assessment.** An osteohistological analysis of the humerus indicates a minimum of 3 years of chronological age for MPC-D 100/553. It is two-thirds the size of the *Yamaceratops* holotype (IGM 100/1315) based on mandibular length (*Makovicky & Norell, 2006*). Some indicators of skeletal immaturity seen in MPC-D 100/553 are long-grained surface texture on the long bones (femur, tibia, fibula, humerus, ulna, and radius) (*Tumarkin-Deratzian, 2009*), the smooth external surface on the postorbital (*Brown & Schlaikjer, 1940*), and open neurocentral sutures of all caudal vertebrae (*Hone et al., 2014*).

### Description and comparisons

**Preservation.** MPC-D 100/553 is reasonably complete and articulated in a life-like crouched position. It is dorsoventrally compressed, with the jugal situated about 1.5 cm below its original position based on the surangular ridge. Sediments around the skeleton are carbonate-cemented, contrasting with the poorly-cemented reddish sediment farther from the skeleton. A series of tectonic-induced joints occur across the skeleton, mainly in the left posterior mandible through the right squamosal, humerus, femur, and tibia. The gap is wider along the right leg and is filled with reddish silt.

**Skull.** The skull (Figs. 4 and 5) is mainly articulated, but the left skull roof and most of the left lateral elements posterior to the orbit are not preserved. The occipital region, braincase, and palatal elements are embedded in the matrix. The skull is proportionately much narrower than the holotype (IGM 100/1315). The basal skull length (from the rostral to quadrate articulation) is 111 mm, and the preserved skull length is 142 mm in the sagittal plane. The width of the skull from the sagittal midline to the lateral tip of the epijugal is 53 mm.

Both narial openings, right and partial left orbit, and both antorbital fossae are preserved. The narial opening is oval and anteroventrally oriented 60° to the horizontal plane. It is anteriorly, ventrally, and posteriorly bounded by the premaxilla and dorsally by the nasal. The ventral margin of the narial opening is lower than the anterior tip of the

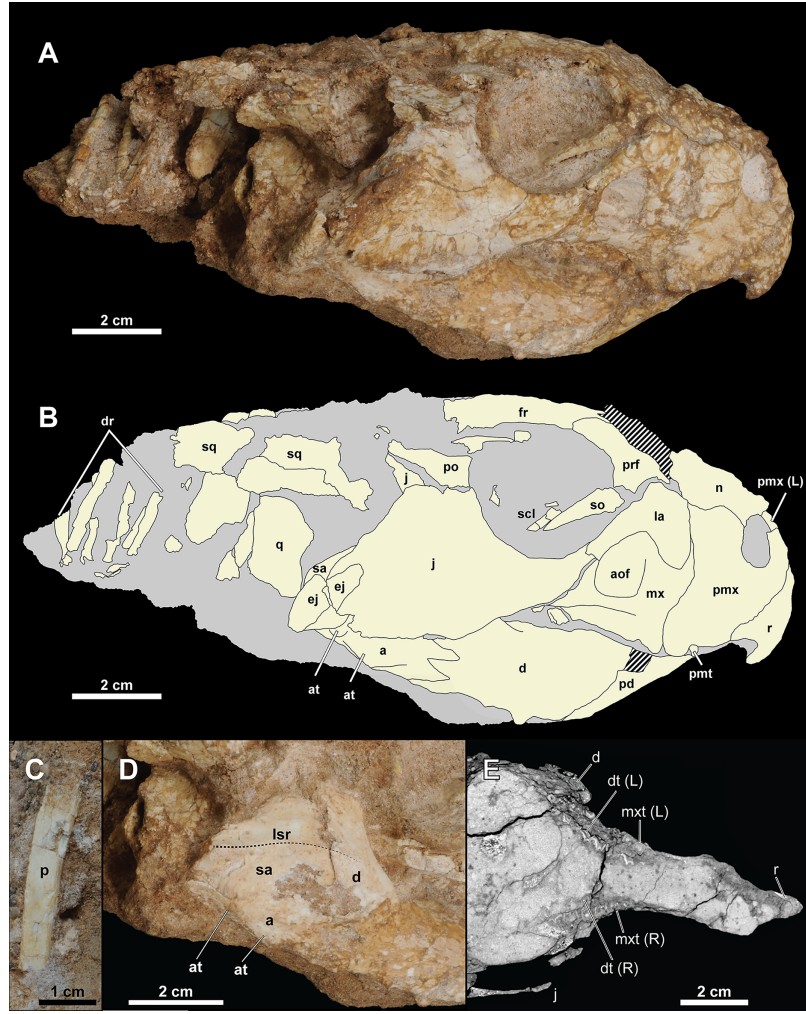

**Figure 4 Skull of *Yamaceratops dorngobiensis* (MPC-D 100/553).** (A) Photograph and (B) interpretive drawing in right lateral view; (C) isolated parietal fragment in dorsal view; (D) posterior mandible in right lateral view; (E) longitudinal micro-CT cross-section of the skull. Abbreviations: a, angular; aof, antorbital fossa; at, angular tubercle; d, dentary; dr, dorsal ribs; dt, dentary tooth; ej, epijugal; fr, frontal; j, jugal; L, bone on the left side; la, lacrimal; lsr, lateral surangular ridge; mx, maxilla; mxt, maxillary tooth; n, nasal; p, isolated parietal; pd, predentary; pmt, premaxillary tooth; pmx, premaxilla; po, postorbital; prf, prefrontal; q, quadrate; R, bone on the right side; r, rostral; sa, surangular; scl, sclerotic ring; so, supraorbital; sq, squamosal.

dorsal process of the rostral and the dorsal margin of the antorbital fossa. The posterodorsal end of the narial opening is about the same level as the anteriormost margin of the orbit.

The orbit is large and circular in lateral view. It is dorsally bounded by the prefrontal, frontal, and postorbital in equal proportions. The ventral margin of the orbit is bounded by the lacrimal and jugal. The lacrimal makes up about 17% of the ventral margin of the orbit, while the jugal bounds the rest. The ratio of the orbit to preorbital length is 0.78 (32 mm to 41 mm), which is between the percentages reported for juvenile *Aquilops*

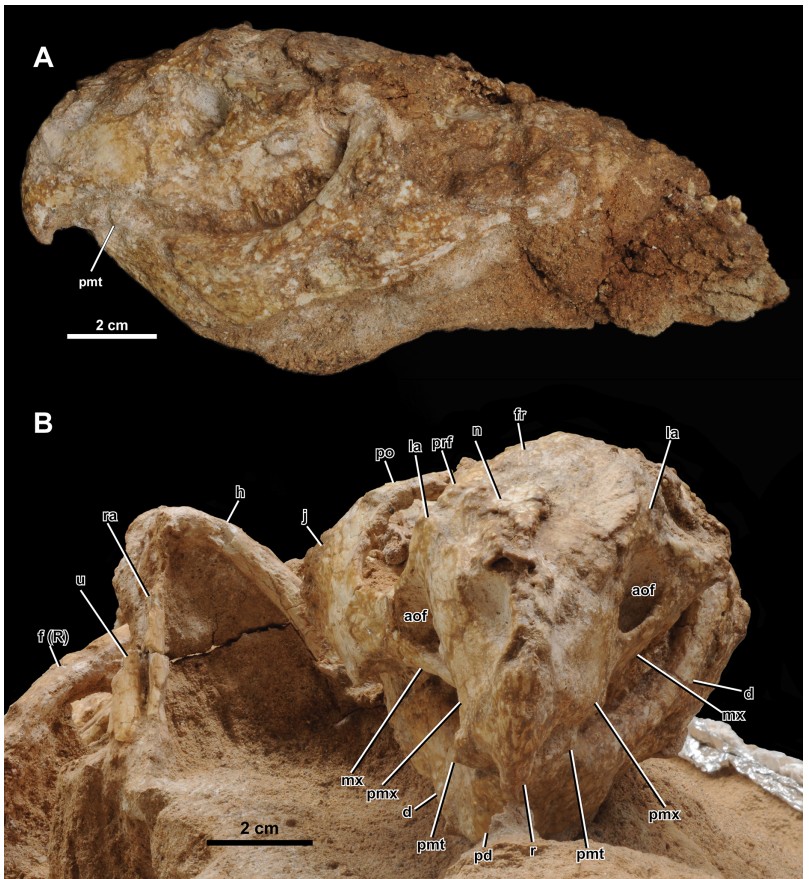

**Figure 5 Skull of *Yamaceratops dorngobiensis* (MPC-D 100/553) in (A) left lateral and (B) anterior view.** Abbreviations: a, angular; aof, antorbital fossa; d, dentary; f (R), right femur; fr, frontal; h, humerus; j, jugal; la, lacrimal; mx, maxilla; n, nasal; pd, predentary; pf, prefrontal; pmt, premaxillary tooth; pmx, premaxilla; po, postorbital; prf, prefrontal; r, rostral; ra, radius; sq, squamosal; sa, surangular; scl, sclerotic ring; so, supraorbital; u, ulna.                               

(OMNH 34557, 0.84), adult *Archaeoceratops* (IVPP V11114, 0.63), and *Auroraceratops* (CAGS-IG-2004-VD-001, 0.64) (*Farke et al., 2014*).

The shape and dimension of the infratemporal fenestra cannot be accurately determined because the bones bounding the fenestra are not in place; the ventral process of the squamosal, which articulates with the quadrate, is missing, and the quadrate is directed medially due to the crushing of the skull.

Given the low curvature of the possible posterior parietal and the squamosal, the supratemporal fenestra would have been triangular. The right squamosal is medially crushed, and the parietal is disarticulated with probable pectoral elements and ribs preserved at the inferred position of the left supratemporal fenestra.

The antorbital fossa is subtriangular in lateral view. It is bounded anteriorly, medially, and ventrally by the maxilla and dorsally by the lacrimal. The fungiform anterior process of the lacrimal does not participate in the antorbital fossa. A deep pocket is ventromedially positioned to the ventral margin of the antorbital fossa ("cleft within antorbital fossa" of *Osmólska, 1986*: 147), as in *Auroraceratops* (*Morschhauser et al.,*

*2018a*), *Bagaceratops* (*Osmólska, 1986*), *Protoceratops andrewsi* (*Brown & Schlaikjer, 1940*), and *Yamaceratops* (*Makovicky & Norell, 2006*). The suture between the maxilla's posterior margin and the lacrimal shaft comprises the posterior margin of the antorbital fossa. The anterior ramus of the jugal contacts the posterior margin of the antorbital fossa but hardly contributes to the rim, as in *Archaeoceratops* (*Sullivan & Xu, 2017*).

Rostral

The anterior margin of the rostral is unkeeled, making a smooth curved edge similar to *Liaoceratops* (*Xu et al., 2002*; but see also *You & Dodson, 2003*), *Mosaiceratops* (*Zheng, Jin & Xu, 2015*), chaoyangsaurids, and psittacosaurids. In lateral view, the ventral process of the rostral becomes lower in height towards the ventral terminus and curves posteroventrally, forming an almost vertical angle with the posteroventral margin of the rostral and the maxillary tooth row, as in *Aquilops* (*Farke et al., 2014*). The width of the ventral process tapers anteriorly but not to the extent of a keel (Fig. 5B). The anterior margin of the ventral process meets at a more acute angle than at the anterior margin of the dorsal process. The lateral surface is slightly convex. The lateral process of the rostral is dorsoventrally low and extends to half-length of the ventral margin of the premaxilla. The dorsal process is slightly expanded posteriorly. The external surface of the rostral is relatively rugose, if not to the extent of anastomosing ridges and grooves as in adult *Yamaceratops* (*Makovicky & Norell, 2006*).

Premaxilla

Both premaxillae are preserved. Each element is subrectangular, higher than long, and anteroposteriorly shorter than the maxilla in the lateral view. A shallow notch is present on the anterodorsal portion of the premaxilla, which forms the anteroventral portion of the narial opening. The ventral margin is slightly convex. The premaxillary-maxillary suture is vertical and dorsally confluent with the posterodorsally inclined premaxillary-lacrimal suture. Although the tip of the posterodorsal process is not fully preserved on both sides, it seems that the premaxilla may have posterodorsally contacted the prefrontal. In palatal view, the premaxilla is mediolaterally widest at the middle.

Maxilla

Both maxillae are well preserved. In lateral view, the maxilla is bounded anterodorsally by the premaxilla, dorsally by the lacrimal, and posteriorly by the jugal. The maxilla bears the whole antorbital fossa with a deep pocket medial to the jugal process. Anterior to the antorbital fossa, the dorsal process receives the expanded anterior process of the lacrimal. The prominent buccal emargination is present along the ventral portion of the jugal process. The emargination converges with the oral margin at the premaxillary-maxillary suture.

Jugal

Only the right jugal is preserved and relatively complete. Only the tip of the dorsal ramus is missing. In lateral view, the jugal is about twice the length of the orbit (66 mm *vs.* 32 mm) and comprises the posteroventral corner of the orbit by more than a quarter.

The ventral margin of the jugal is convex, with a vertex at about half-length between the anterior and posterior rami.

Along the length of the jugal from the tip of the anterior ramus roughly to the base of the dorsal ramus, the suborbital region of the jugal is laterally convex, making a broad rim around the orbit. Dorsal and ventral to this slight curve, the jugal is essentially flat, except for around the dorsolateral ridge on the posterior ramus. The lateral surface of the jugal is textured by shallow grooves that are horizontal on the anterior process and vertical on the posterior process. The posterodorsal ridge along the posterolateral edge of the posterior process is also rugose, likely associated with the epijugal in contact and covered with a keratinous sheath as a jugal horn in life.

The anterior (suborbital) ramus of the jugal is anterodorsally curved and dorsoventrally shallow. The mediolateral width of the anterior ramus increases towards the anterior portion of the jugal. The anterodorsal end of the anterior ramus is forked and overlaps the tip of the ventral ramus of the lacrimal. The anterior ramus meets the jugal process of the maxilla, and the contact extends half its length ventrally. It creates a triple point between the maxilla, jugal, and lacrimal at the posterior vertex of the antorbital fossa.

The posterior (subtemporal) ramus of the jugal is slightly longer and dorsoventrally deeper than the anterior ramus. The jugal with its posterior ramus deeper than the anterior ramus is also in *Aquilops* (*Farke et al., 2014*), *Liaoceratops* (*Xu et al., 2002*), juvenile *P. andrewsi* (*Brown & Schlaikjer, 1940*), juvenile *P. hellenikorhinus* (*Lambert et al., 2001*), holotype of *Yamaceratops* (*Makovicky & Norell, 2006*), and, to a less degree, *Auroraceratops* (*Morschhauser et al., 2018a*). The ramus increases in mediolateral width towards the posterior along its suture with the epijugal. The posterior portion of the posterior ramus gently curves downward, as in *Archaeoceratops* (*You & Dodson, 2003*), *Auroraceratops* (*Morschhauser et al., 2018a*), juvenile *Bagaceratops* (*Czepiński, 2019*), *Beg* (*Yu et al., 2020*), *Liaoceratops* (*Xu et al., 2002*), *Mosaiceratops* (*Zheng, Jin & Xu, 2015*), and juvenile *Protoceratops* (*Brown & Schlaikjer, 1940*; *Lambert et al., 2001*). An elongate epijugal contacts the posterodorsal ridge on the posterodorsal edge of the posterior ramus. The quadratojugal would have medially met the posterior ramus of the jugal. However, the quadratojugal is absent, and the space between the quadrate and the jugal is instead occupied by the surangular, probably due to the dorsoventral crushing of the skull.

The jugal dorsal ramus is much thinner than the anterior and posterior rami. The ventral process of the postorbital laterally covers the anterior part of the dorsal ramus as a scarf joint. The posterior part of the dorsal ramus is a thin sheet of bone that forms the anterodorsal margin of the temporal fenestra. In MPC-D 100/553, although the anterior part of the dorsal ramus contacts the main lateral surface of the jugal, the posterior part of the dorsal ramus is positioned more medially, even more, medial to the anterior part. This condition of the dorsal ramus is comparable to the unique "embayment" or "notch" noted from the same region in the holotype specimen of *Yamaceratops* (*Makovicky & Norell, 2006*: 10). However, this is unlikely to be taxonomically informative and instead interpreted here as a taphonomic artifact due to the postmortem crushing of the skull. The micro-CT images confirmed that the thin sheet of bone is preserved more medially to both the anterior part of the dorsal ramus and the main body of the jugal.

Moreover, it is not in contact with the main body of the jugal but 'floating.' It is, therefore, most parsimonious to interpret this bone as the posterior part of the dorsal ramus that has been broken and displaced. Therefore, the "embayment" is likely due to horizontal breakage and displacement of the dorsal ramus of the jugal following the dorsoventral crushing of the skull in both specimens of *Yamaceratops*.

Epijugal

The epijugal is preserved articulated with the posterodorsal ridge of the right jugal. The jugal-epijugal suture is open. The epijugal is gently crescentic, dorsoventrally low, and anteroposteriorly elongate, spanning the tip of the posterior ramus of the jugal to about halfway up the posterodorsal edge. It does not reach the ventral margin of the jugal, which differs from *Auroraceratops* (*Morschhauser et al., 2018a*), *Bagaceratops* (*Czepiński, 2019*), '*Graciliceratops*' (*Sereno, 2000*), both juvenile and adult *P. andrewsi* (*Brown & Schlaikjer, 1940*), and the holotype of *Yamaceratops* (*Makovicky & Norell, 2006*). The transverse section of the epijugal is triangular, and its contact with the jugal is concave. The posterolateral tip of the epijugal is rounded. The lateral surface is highly textured, suggesting it was covered with keratin in life as a jugal horn. The posterodorsal position and elongate shape of the epijugal in MPC-D 100/553 are similar to *Leptoceratops*, *Montanoceratops*, and *Yamaceratops* (*Makovicky, 2010*).

Nasal

The right nasal is crushed, and the surface is poorly preserved. The nasal comprises the dorsal border of the narial opening and contacts the premaxilla, lacrimal, prefrontal, and frontal. The nasal is flat and mediolaterally wide relative to the frontal. The mediolaterally widest part of the nasal contacts the lacrimal and is almost as wide as the frontal at the contact with the postorbital on the orbital margin (15 mm *vs.* 16 mm).

Lacrimal

Both left and right lacrimals are preserved. The lacrimal is shaped like a hatchet, with a narrow ventral ramus and greatly expanded anterior ramus. The anterior ramus of the lacrimal is fungiform in lateral view, similar to *Auroraceratops* (*You et al., 2005*), but unlike the rectangular shape in *Beg* (*Yu et al., 2020*). It is surrounded by the maxilla ventrally and meets the premaxilla anterodorsally. Although the point of contact is crushed, the anterior ramus seems to contact the nasal dorsally as in *Archaeoceratops* (*You & Dodson, 2003*). This condition differs from *Aquilops* (*Farke et al., 2014*) and *Auroraceratops* (*You et al., 2005*), which have no contact between the lacrimal and the nasal, and also *Bagaceratops* (*Maryańska & Osmólska, 1975*), *Leptoceratops* (*Sternberg, 1951*), and *Protoceratops* (*Brown & Schlaikjer, 1940*; *Lambert et al., 2001*), which have an extensive contact between the two bones.

The contact between the premaxilla, nasal, lacrimal, and prefrontal forms a quadruple junction. The anterior ramus of the lacrimal does not contribute to the antorbital fossa. The ventral ramus of the lacrimal is rod-like, similar to *Aquilops* (*Farke et al., 2014*), *Archaeoceratops* (*You & Dodson, 2003*), *Auroraceratops* (*You et al., 2005*), *Leptoceratops* (*Sternberg, 1951*), and *Liaoceratops* (*You, Tanoue & Dodson, 2007*). The ventral ramus

 

contributes to the anteroventral rim of the orbit and constitutes the posterodorsal margin of the antorbital fossa but does not contribute to the medial wall of the antorbital fossa extensively as in protoceratopsids (*Brown & Schlaikjer, 1940*; *Czepiński, 2019*). The lacrimal also articulates with the supraorbital, posterior to the posterodorsal contact with the prefrontal.

Prefrontal

The prefrontal is small and relatively thick, contributing to about one-third of the dorsal half of the orbit. The ventral process is posterior to the lacrimal and articulates with the supraorbital. The prefrontal differs from *Bagaceratops* (*Czepiński, 2019*), *P. andrewsi* (*Brown & Schlaikjer, 1940*), and adult *Yamaceratops* (*Makovicky & Norell, 2006*), which have an extensive ventral process.

Supraorbital

We use the term supraorbital instead of "palpebral" because the former is homologous in ornithischian dinosaurs, and the latter originally refers to a metaplastic ossification in crocodilians (*Maidment & Porro, 2010*; *Nesbitt, Turner & Weinbaum, 2012*). Only the right supraorbital is wholly preserved. It is large, with its length reaching half of the orbit. It is elongated and triangular. The supraorbital is articulated with the prefrontal and lacrimal at the base. However, it is collapsed with the tip pointing medioventrally into the orbit, ventral to the collapsed sclerotic ring.

Frontal

The frontal is partially preserved on the right side of the skull. The posterior part, including the frontal fossae and the suture with the parietal, is not preserved. The frontal is flat dorsal to the orbit and gently slopes anteriorly in its contact surface with the nasal. The margin and striations for the contact with the nasal are preserved. In dorsal view, the posteriormost contact with the nasal would have been anterior to the contact with the prefrontal.

Postorbital

The right postorbital is triradiate with the three extremities damaged. The anterior and posterior processes are poorly preserved due to compression, and the narrower ventral process is demarcated by a layer of sediment left in the jugal. The dorsal surface of the postorbital is damaged, and only the anterior margin can be traced. The ventral process would have been narrow and pointed anteroventrally, based on the depression left on the jugal. In the posterodorsal corner of the orbit, the postorbital is divided into the dorsal and lateral surfaces without a distinct ridge. The anterior process at the dorsal surface bounds the posterodorsal corner of the orbit and meets the frontal. The posterior contact with the squamosal is indiscernible and probably damaged, but the preserved length of the postorbital is 33 mm, almost equal to the length of the orbit. The lateral surface of the postorbital is flat and smooth. It is excluded from the infratemporal fenestra by the thin and broad dorsal process of the jugal. The posterior margin of the postorbital is slightly concave and converges with the anterior dorsal process of the jugal.
Squamosal

Only the right squamosal is preserved. Its ventral ramus is damaged, and the contact region with the postorbital, quadrate, and parietal are missing. In lateral view, the dorsoventral height of the anterior process is highest in the preserved anterior margin. The squamosal tapers towards the posterior margin, and its lateral surface is flat to slightly concave. In dorsal view, the squamosal is gently convex laterally, although the transverse width is greatest at the anterior margin due to the lateral crushing of the frill.

Parietal

An isolated element located anteroventral to the right distal fibula is tentatively referred to as the partial posterior margin of the parietal (Fig. 4C). It is long, narrow, somewhat flat, thin, and slightly curved. It has a low ridge at the dorsal surface, along the concave anterior margin. Thin, narrow, and only slightly curved posterior parietals indicate the presence of large parietal fenestrae. Among non-ceratopsid ceratopsians, this character is only known in *Cerasinops* (*Chinnery & Horner, 2007*), '*Graciliceratops*' (*Sereno, 2000*), and juveniles of the protoceratopsid *P. andrewsi* (*Brown & Schlaikjer, 1940*; *Handa, Watabe & Tsogtbaatar, 2012*) and *Bagaceratops* (*Czepiński, 2019*). Other basal neoceratopsians with parietal fenestra preserved are *Liaoceratops* (*Xu et al., 2002*) and *Auroraceratops* (*Morschhauser et al., 2018a*). The posterior parietals are not thin in these taxa, and the parietal fenestra is small. The "very slender median and posterior parietal frill margin" similar to MPC-D 100/553 was suggested as a unique character of '*Graciliceratops*' by *Sereno (2000*: 491) but is known to be ontogenetically variable in *P. andrewsi* (*Brown & Schlaikjer, 1940*; *Handa, Watabe & Tsogtbaatar, 2012*; as noted by *Makovicky & Norell, 2006*) and *Bagaceratops* (*Czepiński, 2019*).

Quadrate

The right quadrate is preserved. The condyles and their contact with the articular are obscure. The quadrate head is turned medially, and the pterygoid wing is facing laterally, probably due to the deformation of the skull. There is a straight ridge along the posterolateral edge of the shaft, which is slightly pointed at mid-height.

Braincase

Much of the braincase is incased in the matrix, and detailed analyses from CT data will be presented elsewhere.

Sclerotic ring

A portion of the right sclerotic ring is exposed, lying on top of the tip of the supraorbital. Five partial plates can be seen, of which the anteroventral three are articulated, with the posteroventral ones overlapping the anterodorsal ones.

**Mandible.** The right mandible is completely preserved except for around the mandibular fenestra. The surangular is missing in the left mandible, and much of the lateral surface is broken.
Predentary

The predentary has a sharp tip and an anteroposteriorly narrow oral margin. The anterior margin is straight and gently keeled. Posteroventrally, the ventral process splits about ventral to the anteroventral corner of the posterior process, forming an obtuse angle in lateral view (Figs. 5A, Fig. S1), and the ends of each side form another angle abruptly at the level of the ventral margin of the dentary. The external surface of the predentary is textured with low grooves and ridges longitudinally radiating from the tip.

Dentary

Both sides of the dentary are preserved, and it comprises 59% of the mandible length (54 mm out of 92 mm) in lateral view. The ventral margin of the dentary is straight. The dorsal tip of the coronoid process is textured with anteroventrally directed grooves. The dorsal margin of the coronoid process is higher than the dorsal margin of the surangular. The buccal emargination is prominent. The lateral surface of the dentary ventral to the buccal margin is flat and vertical with shallow striations. This condition differs from the laterally flared and striated dentary surface of the holotype of *Yamaceratops* (*Makovicky & Norell, 2006*).

Surangular

The right surangular is almost complete except for the anteroventral corner. The surangular occurs above the angular and behind the dentary. The dorsal margin of the surangular is sinuous with convex anterior, concave mid-portion, and convex posterior margins. The surangular has a pronounced lateral ridge that is almost horizontal and only slightly inclined posteroventrally. It is laterally highest in the posterior margin and inclines anteriorly. The lateral ridge extends onto the dentary, decreasing in height. The posteroventral margin of the lateral surface of the surangular is ventrally convex, almost touching the posterior angular tubercle.

Among *Yamaceratops*, the lateral surangular ridge is lower in MPC-D 100/553 and higher in the holotype IGM 100/1315. It should be noted that the lateral surangular ridge is well-developed in *Archaeoceratops*, *Bagaceratops*, *Beg*, *Cerasinops*, *Leptoceratops*, *Protoceratops*, and *Yamaceratops* (*Makovicky & Norell, 2006*; *Tanoue, You & Dodson, 2010*; *Yu et al., 2020*) but very low to absent in *Auroraceratops*, *Liaoceratops*, *Prenoceratops*, *Udanoceratops*, and ceratopsids (*Chinnery, 2004a*; *Makovicky & Norell, 2006*; *Morschhauser et al., 2018a*).

Angular

The right angular is almost complete, and the left angular is missing the posterior half of the dorsal and lateroventral portion. It is bifurcated anteriorly along the suture with the dentary and covered by the surangular dorsolaterally. The dorsal margin of the angular is very thin mediolaterally. The lateral surface of the angular is flat and vertical except for the posteriormost part of the angular where the lateral surface faces posteroventrally. The right angular has ventral and posterior tubercles. The ventral tubercle is located dorsolateral to the ventral margin at about mid-length of the angular. The posterior tubercle is located just ventral to the surangular at the posterior quarter. The ventral

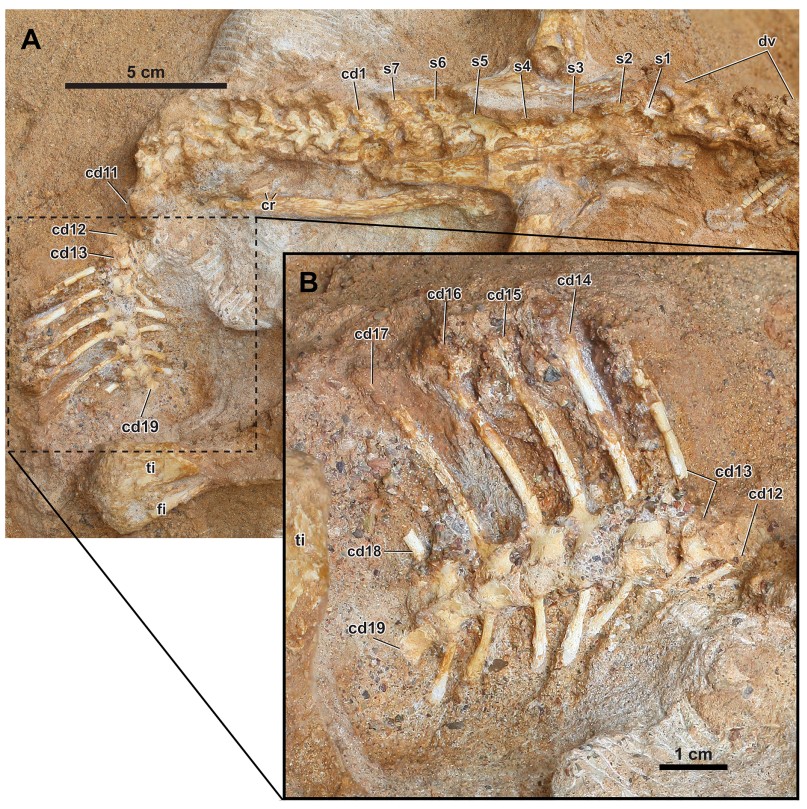

**Figure 6 Axial skeleton of *Yamaceratops dorngobiensis* (MPC-D 100/553).** (A) Dorsal to anterior caudal vertebrae in right lateral view; (B) middle caudal vertebrae with very tall neural spines in right lateral view. The s1 is the dorsosacral, s2–5 are the true sacrals, and s6–7 are the caudosacrals. Abbreviations: cd, caudal vertebra; cr, caudal ribs; dv, dorsal vertebrae; s: sacral vertebra; ti, tibia; is, ischium.

tubercle is low and laterally flat, whereas the posterior tubercle is low and domed. The margin of the mandibular fenestra is damaged but located anterodorsal to the angular.

**Dentition.** In lateral view, all dentary teeth are covered laterally by the maxillary teeth, and the ventral edges of the maxillary teeth cannot be observed due to the dorsoventral crushing of the skull. However, from the articulated and well-preserved nature of the skull, most, if not all, teeth are likely present, as preliminarily seen from the CT images (Fig. 4D). The cingulum is developed on the left maxillary teeth (Figs. 5A, Fig. S2) as in the other referred *Yamaceratops* specimen IGM 100/1303 (*Jin et al., 2009*; *Makovicky & Norell, 2006*; figs. 5, 15). Judging from the CT data, there seem to be two premaxillary, nine maxillary, and eight dentary teeth with well-developed primary ridges. There is a second row of replacement teeth.

**Overview of the axial elements.** The axial column is fully articulated (Figs. 2 and 6). The cervical series and most of the dorsal vertebrae are lost due to erosion. The posterior dorsal vertebrae are poorly preserved. Although the contacts are difficult to interpret, the sacral vertebrae are mostly complete. The anterior caudal vertebrae are in place, although the right transverse processes, right half of the centra, and proximal chevrons are

missing. In the transition from the anterior to middle caudal vertebrae, the tail is abruptly turned anteroventrally, with the caudal series still in ascending order with wide spacing. The matrix obscures the posterior caudal vertebrae, but they probably lie ventral to the right tibia and fibula.

Dorsal vertebrae and dorsal ribs

The posterior dorsal vertebrae are articulated but in poor condition, and their neural spines are lower than those of the anterior caudal vertebrae. The exact morphology of the dorsal vertebrae is hard to observe, as they are fragmented and therefore not fully prepared to maintain stability. The anterior and posterior dorsal ribs are preserved without the proximal portions. The ribs on the right side of the body were probably partially articulated from their respective vertebrae, given they are mostly parallel in the series. The rib cage has collapsed and lies on its left side so that the left ribs lie just below the right ribs. The ribs are long and slender, and the cross-section is oval in the preserved portions.

Sacral vertebrae and sacral ribs

Seven sacral vertebrae are present between the ilia (s1–7). The sacrum is composed of one dorsosacral vertebra (s1), four true sacral vertebrae (s2–s5), and two caudosacral vertebrae (s6–s7). Of the four true sacral vertebrae, s2–4 are medial to the acetabulum. The neural spines of the sacral vertebrae are tilted left, although to a lesser degree in sacral vertebrae 2–5, so that the right side is primarily exposed in dorsal view. The left side is covered in the matrix, roofed by the neural spines and left ilium. Whether the centra of s1–5 are co-ossified cannot be observed, and the centra of s5–7 are not co-ossified. The pre- and postzygapophyses of s1–7 are poorly developed but not co-ossified. The sacral neural spines are low and anteroposteriorly expanded, widest in s2–5, and the anterior margin is sloping in s6–7. The neural spines are not co-ossified, although the neural spines of s3–4 are in close proximity. From the slight displacement and tilting of both ilia, it can be inferred that neither the distal ends of the transverse processes nor the sacral ribs are co-ossified with the ilium. The distal ends of the transverse processes are covered on top by the tilted ilium on the right side. However, it is unlikely that the distal ends are co-ossified from the low anteroposterior expansion and wide spacing in the proximal portion. The posterior sacral vertebrae are poorly preserved, and the respective sacral ribs are not preserved.

Caudal vertebrae, caudal ribs, and chevrons

The anterior to middle caudal vertebrae (cd1–19) are preserved in articulation (Fig. 6A). The more posterior caudal vertebrae are not visible due to the overlying right tibia. There is a slight displacement between the eleventh and twelfth caudal vertebrae following the abrupt turn of the caudal series (Fig. 6A). All neurocentral sutures are open. In the anteriormost caudal vertebra (cd1), which is located between the posterior ends of the ilia, the anteroposterior width of the neural spine and the length of the centrum is intermediate between the last caudosacral vertebra (s7) and the second anterior caudal vertebra (cd2). The overlap of zygapophyses is more extensive at the anterior than the

middle caudal vertebrae. Overall, the neural spines of the anterior caudal vertebrae are slightly taller and narrower anteroposteriorly than those from the dorsal vertebrae.

The middle caudal vertebrae possess the highest neural spines (preserved in cd13–17) that are much taller and anteroposteriorly narrower than those of the anterior caudal vertebrae (Fig. 6B). In the mid-caudal vertebrae cd17, the height of the preserved neural spine (32 mm) is more than five times the height of its corresponding posterior centrum face (6 mm). It cannot be determined whether the tallest neural spine is at the middle portion of the tail or the more posterior part because the more posterior part of the tail is covered by the overlying matrix and the right leg. The neural spines are gently curved posteriorly, with the base nearly vertical to the centra. The neural spines are only slightly expanded distally. The cross-section of the neural spines is oval. The zygapophyses meet nearly horizontally, and each neural arch is tall relative to the centrum. The centra of the middle caudal vertebrae are slightly spool-shaped (waisted), with a concave ventral margin in lateral view. The distal ends of the caudal ribs for the anterior caudal vertebrae are preserved, meeting the ischium (Figs. 2 and 6A). They are anteroposteriorly wide and rectangular, with parallel anterior and posterior margins directed laterally rather than posterolaterally. Caudal ribs are not preserved for the middle caudal vertebrae. The facets for articulation with the caudal ribs are present in cd18, and their presence in more posterior caudal vertebrae cannot be verified. Very tall middle caudal neural spines that are more than four times the height of the centrum occur in *Koreaceratops*, *Montanoceratops*, and *P. andrewsi* (*Tereshchenko, 2008*; *Lee, Ryan & Kobayashi, 2011*; *Tereshchenko & Singer, 2013*). The case for *Bagaceratops* should be taken with caution because the specimen PIN 3143/11, a partial tail, that was referred to and provided data for this taxon (*e.g., Tereshchenko, 2007, 2008*; *Tereshchenko & Singer, 2013*) likely instead belongs to *P. andrewsi* (*Czepiński, 2020*).

The chevrons of cd11–18 are preserved. The chevrons are rod-like and elongated but much shorter and a bit narrower proximally than the corresponding neural spines. They are wider distally, as in *Leptoceratops* and *P. andrewsi*, but not as much expanded as in *Auroraceratops* and *Koreaceratops* (*Morschhauser et al., 2018b*). The chevron is nearly twice as long as the height of the respective centrum face in cd17 (11 mm and 6 mm, respectively).

Ossified tendons

Three ossified tendons are partially preserved on the right dorsal side of the two anteriormost sacral vertebrae (s1–2). The ossified tendons are parallel to each other and do not overlap. Neither the presence nor the extent of ossified tendons is taxonomically diagnostic because they are subjective to taphonomic biases even among a single basal neoceratopsian taxon from a "geographically and stratigraphically restricted area" (*Morschhauser et al., 2018b*: 111). It has been shown that the ossified tendons in hadrosaurine dinosaurs (*Maiasaura peeblesorum* and *Brachylophosaurus canadensis*) were not induced by biomechanical stresses (non-pathologic), and their development began early, as they were even found from a nestling specimen (*Adams & Organ, 2005*).

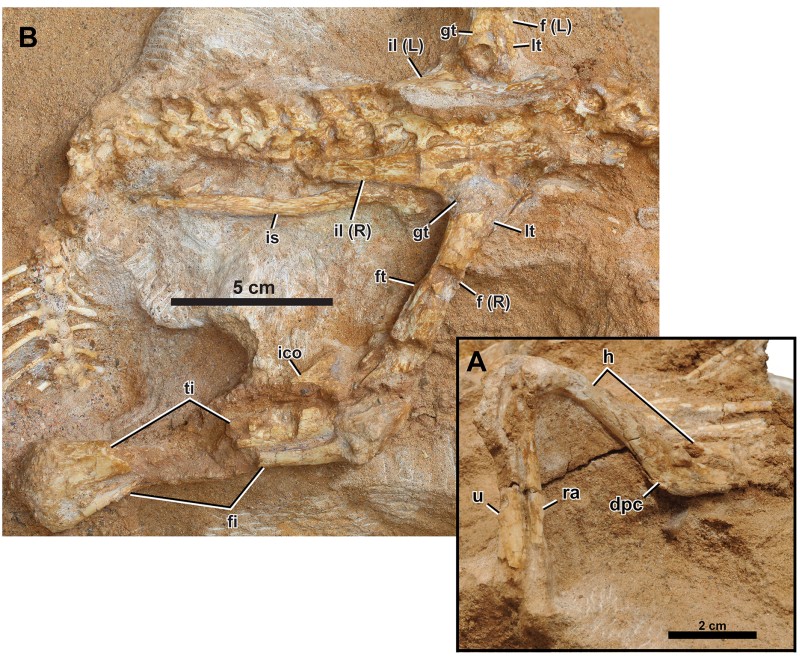

**Figure 7 Right appendicular skeleton of *Yamaceratops dorngobiensis* (MPC-D 100/553).** (A) Fore-limb in ventrolateral view; (B) pelvic girdle and hind limb in lateral view. Abbreviations: dpc, delto-pectoral crest; f, femur; fi, fibula; ft, fourth trochanter; gt, greater trochanter; h, humerus; ra, radius; u, ulna. L, bones on the left side; lt, lesser trochanter; R, bones on the right side; ti, tibia; ico, inner condyle of the tibia; il, ilium; is, ischium.

**Overview of the appendicular elements.** Several authors have recently done descriptive and comparative work on the basal ceratopsian appendicular elements for several taxa: they are well described and compared in *Auroraceratops* (*Morschhauser et al., 2018b*), *Ferrisaurus* (*Arbour & Evans, 2019*), *Ischioceratops* (*He et al., 2015*), and *Yinlong* (*Han et al., 2018*). Juvenile specimens of *Montanoceratops* (*Chinnery & Weishampel, 1998*) and *Psittacosaurus lujiatunensis* (*Hedrick et al., 2014*) help to clarify the ontogeny of basal ceratopsian appendicular elements. This pattern was particularly well described for *P. andrewsi* (*Słowiak, Tereshchenko & Fostowicz-Frelik, 2019*) and altogether formed the foundation for comparative work between taxa and ontogenetic stages of specific taxa.

**Forelimb.** The right humerus, radius, and ulna are preserved articulated (Fig. 7A), although the proximal ulna is missing and the glenoid of the radius is indiscernible.

Humerus

The right humerus is preserved, with a crack across the proximal end of the deltopectoral crest to near the base of the shaft. Overlapping dorsal ribs obscure the humeral head. The humerus is gracile, with less expanded proximal and distal ends and a slender shaft than juvenile *P. andrewsi* and other basal ceratopsian specimens of more advanced growth stages (*Słowiak, Tereshchenko & Fostowicz-Frelik, 2019*). It is most similar to '*Graciliceratops*' (ZPAL MgD-I/156). The shaft of the humerus is oval in cross-section as the proximally positioned deltopectoral crest does not extend down to the shaft as a ridge. In *P. andrewsi*, the cross-section of the humeral shaft is triangular in

juveniles (*Brown & Schlaikjer, 1940*) and distally suboval in young adults (*Słowiak, Tereshchenko & Fostowicz-Frelik, 2019*). The shaft is slightly bowed posteriorly in lateral view, as in all basal neoceratopsians except *Cerasinops* (*Słowiak, Tereshchenko & Fostowicz-Frelik, 2019*). The deltopectoral crest is low in lateral view and transversely flat. The overall development of the deltopectoral crest is more similar to *Cerasinops* than *Auroraceratops* and *P. andrewsi*. In posterior view, the ulnar condyle extends more distally and is transversely narrower than the radial condyle.

Ulna

The distal half of the right ulna is preserved. The cross-section of the ulnar shaft is widely ovate (*i.e.*, roughly oval with the anterior side wider), and the distal end of the ulna is mediolaterally narrow. The distal shaft of the ulna is straight in posterior view, as the general configuration of basal neoceratopsians (*Chinnery & Horner, 2007*) and in contrast to the medially-bending condition in some leptoceratopsids (*Cerasinops* MOR 300, *Ferrisaurus* RBCM P900, *Prenoceratops* TCM 2003.1.8, and *Udanoceratops* PIN 3907/11) (*Chinnery, 2004b*; *Chinnery & Horner, 2007*; *Arbour & Evans, 2019*).

Radius

The right radius is preserved except for the proximal end. The mid-shaft is bluntly triangular in cross-section as in *P. andrewsi* and *Auroraceratops* (*Słowiak, Tereshchenko & Fostowicz-Frelik, 2019*). The lateral margin of the radius is straight.

**Pelvic Girdle.** The pelvic girdle is well preserved, other than the effects of dorsoventral compression during diagenesis.

Ilium

Both ilia are well preserved (Figs. 7B and 8A), only missing the thin anteriormost portions. The pubic peduncles and ischiadic peduncles are partly covered in the matrix. The overall morphology corresponds to the left ilium of IGM 100/1303, referred to as *Yamaceratops*, 62% larger than that of MPC-D 100/553 (*Makovicky & Norell, 2006*). In lateral view, the dorsal margin of the ilium is generally convex. The posterior end of the postacetabular process points slightly dorsally and resembles the condition of *Mosaiceratops* (*contra Zheng, Jin & Xu, 2015*) and *P. andrewsi* (*Słowiak, Tereshchenko & Fostowicz-Frelik, 2019*). The lateral surface of the ilium is smooth. In dorsal view, the dorsal margin of the ilium shows a weak sigmoid curvature where the preacetabular process is slightly laterally deflected, and the postacetabular process is slightly convex laterally. The dorsal margin of the ilium is only slightly everted laterally.

The preacetabular processes are very short relative to the postacetabular processes. The postacetabular process of *Yamaceratops* is relatively more elongate than that of most other basal ceratopsians, only rivaled by *Psittacosaurus xinjiangensis* (IVPP V7698; IVPP V7701) and adult *P. andrewsi* (*e.g.*, AMNH 6424) (*Sereno & Chao, 1988*; *Słowiak, Tereshchenko & Fostowicz-Frelik, 2019*). The postacetabular process is dorsoventrally slightly taller than the preacetabular process. The postacetabular process tapers posteriorly, although this varies ontogenetically or individually in *Yinlong* (*Han et al., 2018*).

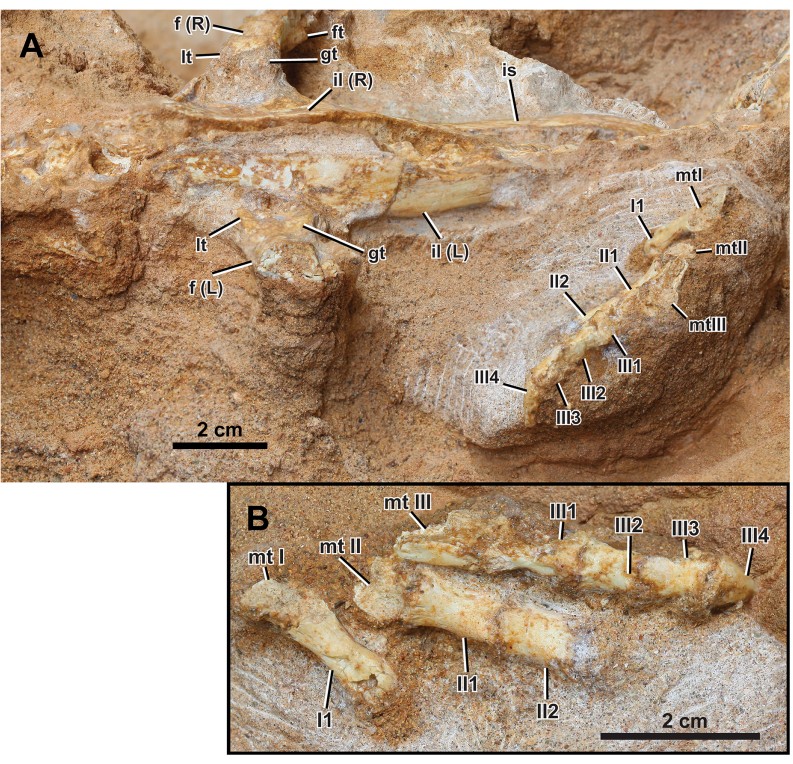

**Figure 8 Left appendicular skeleton of *Yamaceratops dorngobiensis* (MPC-D 100/553).** (A) Ilium and pes in lateral view; (B) pes in dorsal view. Abbreviations: f, femur; ft, fourth trochanter; gt, greater trochanter; L, bones on the left side; lt, lesser trochanter; mt, metatarsal; R, bones on the right side; il, ilium; I, pedal digit I; II, pedal digit II; III, pedal digit III.               

The lateral side of the postacetabular process is slightly longitudinally concave, and the brevis shelf is narrow.

The ilium is low above the acetabulum, with the depth of the ilium only half that of the length between the base of the pre- and postacetabular processes. It is unlike other members of Cerapoda but similar to the basal pachycephalosaur (*Dieudonné et al., 2020*; 'heterodontosaurid'), *Abrictosaurus*, and the non-neoceratopsian ceratopsians *Stenopelix* and *Yinlong* (*Han et al., 2018*). The ilium is relatively deeper in the larger specimen of *Yamaceratops* (IGM 100/1303), so this character is ontogenetically variable in *Yamaceratops*.

Ischium

The right ischium is exposed, with its medial side facing up, as in dorsoventrally compressed specimens of *Leptoceratops* (CMN 8887; *Sternberg, 1951*) and juvenile *Protoceratops* (MPC-D 100/530; *Fastovsky et al., 2011*) (Fig. 7B). The overlying ilium obscures the end of the pubic peduncle and the iliac peduncle. The ischium is long, dorsoventrally slender, straight, and lacks an obturator process. The ischial shaft is transversely flat, and its medial surface is smooth. The ischial shaft is laterally compressed as in leptoceratopsids such as *Montanoceratops* and *Leptoceratops*, contrary to other basal neoceratopsians whose cross-section is oval (*e.g.*, *Auroraceratops*, *Koreaceratops*,
*Mosaiceratops*, and *P. andrewsi*) (*Słowiak, Tereshchenko & Fostowicz-Frelik, 2019*). The ischium is slightly laterally convex, as in *Mosaiceratops* and *P. andrewsi*, unlike the straight ischial shaft in *Archaeoceratops* and *Koreaceratops* (*Słowiak, Tereshchenko & Fostowicz-Frelik, 2019*). The morphology of the ischial shaft is unlikely to be due to postmortem deformation, as the bone surface shows no sign of breakage from flattening. The distal end is covered in the matrix and lies beneath the caudal vertebrae.

**Hind Limb.** Both hind limbs are partially preserved (Figs. 2, 7B, and 8). The right hind limb is preserved articulated and somewhat extended, although the parts distal to the tarsus are absent. The preserved portion of the left hind limb comprises the proximal femur articulated with the ischium and part of the pes.

Femur

The right femur and the proximal third of the left femur are preserved (Figs. 2, 7B, and 8A). Both femora are articulated with the ilium, so the femoral head cannot be seen. The proximal portion of the left femur is only slightly expanded anteroposteriorly. The lesser and greater trochanters are divided by a narrow and shallow groove, although this may have been deeper on the poorly preserved proximal ends. The femoral shaft is straight and slender in lateral view. This condition is ontogenetically variable in *P. andrewsi*, as the femora are arched in very small juveniles but straight in small to large adults (*Słowiak, Tereshchenko & Fostowicz-Frelik, 2019*). Only the base of the fourth trochanter is preserved. It is located just proximal to the middle of the femur, facing posteromedially. The preservation of the fourth trochanter is known to vary with taphonomy among specimens of *Psittacosaurus lujiatunensis* (*Hedrick et al., 2014*; *Persons & Currie, 2020*). The matrix obscures the inner and outer condyles.

Tibia

Only the right tibia is preserved and is articulated with the fibula and femur (Figs. 2 and 7B). Its posterior surface is exposed in dorsal view. The middle portion of the shaft is missing. The tibia is long and slender, with only modestly expanded ends. At the proximal end, the tectonic joint cuts the inner condyle away. The distal end of the tibia is laterally expanded to meet the fibula. The distal end of the tibia is not angled medially and is exceptional among non-ceratopsid ceratopsians, with *Auroraceratops* and *Ischioceratops* (*Słowiak, Tereshchenko & Fostowicz-Frelik, 2019*).

Fibula

Although the middle portion of the shaft is missing, the right fibula is articulated with the tibia. The proximal end is anteroposteriorly expanded and mediolaterally compressed. The distal fibula is narrow with an oval cross-section. Its distal end curves anteriorly to meet the anterior surface of the distal tibia.

Pes

Only the left pes is preserved and articulated (Fig. 8). The preserved elements are digits I to III, with only the distal portions of metatarsals I to III preserved in articulation. The pedal ungual is preserved in digit III. The surfaces are broken proximolaterally on

phalanx I-1, laterally on phalanges I-1 and -2, and dorsally on metatarsals I to III. The first phalanx of digit I is longer than that of digit II, similar to that of digit III in length. The phalanx of digit I is narrower than those of digits II and III. The second phalanx of digit II is shorter than the first but longer than the second phalanx of digit III. The phalanx III-2 is subequal in length but slightly shorter than III-3.

As in MPC-D 100/553, the length of phalanx II-1 is more prolonged than II-2 in *Auroraceratops* (*Morschhauser et al., 2018b*), *Cerasinops* (*Gilmore, 1939*; *Chinnery & Horner, 2007*), 'Graciliceratops' (*Maryańska & Osmólska, 1975*), *Koreaceratops* (*Lee, Ryan & Kobayashi, 2011*), *Leptoceratops* (*Sternberg, 1951*), *Montanoceratops* (*Chinnery & Weishampel, 1998*), *P. andrewsi* (*Brown & Schlaikjer, 1940*), *Psittacosaurus amitabha* (*Napoli et al., 2019*), and indeterminate leptoceratopsid PIN no. 4046/11 (*Tereshchenko, 2008*; *Słowiak, Tereshchenko & Fostowicz-Frelik, 2019*). However, phalanx II-1 is shorter than II-2 in *Archaeoceratops* (*You & Dodson, 2003*) and *Yinlong* (*Han et al., 2018*).

As in MPC-D 100/553, the length of phalanx III-2 is subequal to III-3 in *Ferrisaurus* (*Arbour & Evans, 2019*), 'Graciliceratops' (*Maryańska & Osmólska, 1975*), *Koreaceratops* (*Lee, Ryan & Kobayashi, 2011*), *P. andrewsi* (*Brown & Schlaikjer, 1940*), and an indeterminate leptoceratopsid PIN 4046/11 (*Tereshchenko, 2008*; *Słowiak, Tereshchenko & Fostowicz-Frelik, 2019*; *Arbour & Evans, 2019*). The III-2 are longer than or subequal in length to III-3 with variable relative lengths in specimens of *Auroraceratops* (*Morschhauser et al., 2018b*), *Cerasinops*, and *Leptoceratops* (*Arbour & Evans, 2019*).

The morphology of the pedal ungual III is intermediate between hoof-like and claw-like, although its medial and lateral edges and the tip are broken (Fig. S3). The proximal end of the pedal ungual is about the same width as the distal end of the preceding pedal phalanx. The relative width of the pedal ungual is known to be ontogenetically variable in *P. andrewsi* (*Słowiak, Tereshchenko & Fostowicz-Frelik, 2019*) and even in the ceratopsid *Chasmosaurus belli* (*Currie et al., 2016*): pedal unguals are relatively longer and narrower in small juveniles than adult individuals. The pedal unguals of two ontogenetic stages of *Yamaceratops* (MPC-D 100/553; IGM 100/1303) are not as elongate as those in some leptoceratopsids (*Cerasinops*, *Ferrisaurus*, *Leptoceratops*), nor are they as broad as those of adult *P. andrewsi* (*Słowiak, Tereshchenko & Fostowicz-Frelik, 2019*; *Arbour & Evans, 2019*).

## Bone histology

The humerus of MPC-D 100/553 was examined for histological study (Fig. 9). The tissue is mainly fibro-lamellar and overall similar to subadult *Protoceratops andrewsi* (*Fostowicz-Frelik & Słowiak, 2018*). The cortex of the humerus exhibits longitudinal, reticular, and plexiform vascularization. Four zones are observed. The endosteal region contains a small amount of trabecular bone and plexiform bone matrix with large bone resorption cavities. A small number of radial vessels are observed. Bone vascularity shifts from longitudinal to laminar toward the periosteal region. The lacunae are less dense in the outer two zones than inside. A total of three lines of arrested growth (LAGs) are preserved. The spacing between the second and third LAGs is shorter than the first and second LAGs, as in a fibula of *Psittacosaurus lujiatunensis* IVPP V14341.1 with three LAGs (*Zhao et al.,*

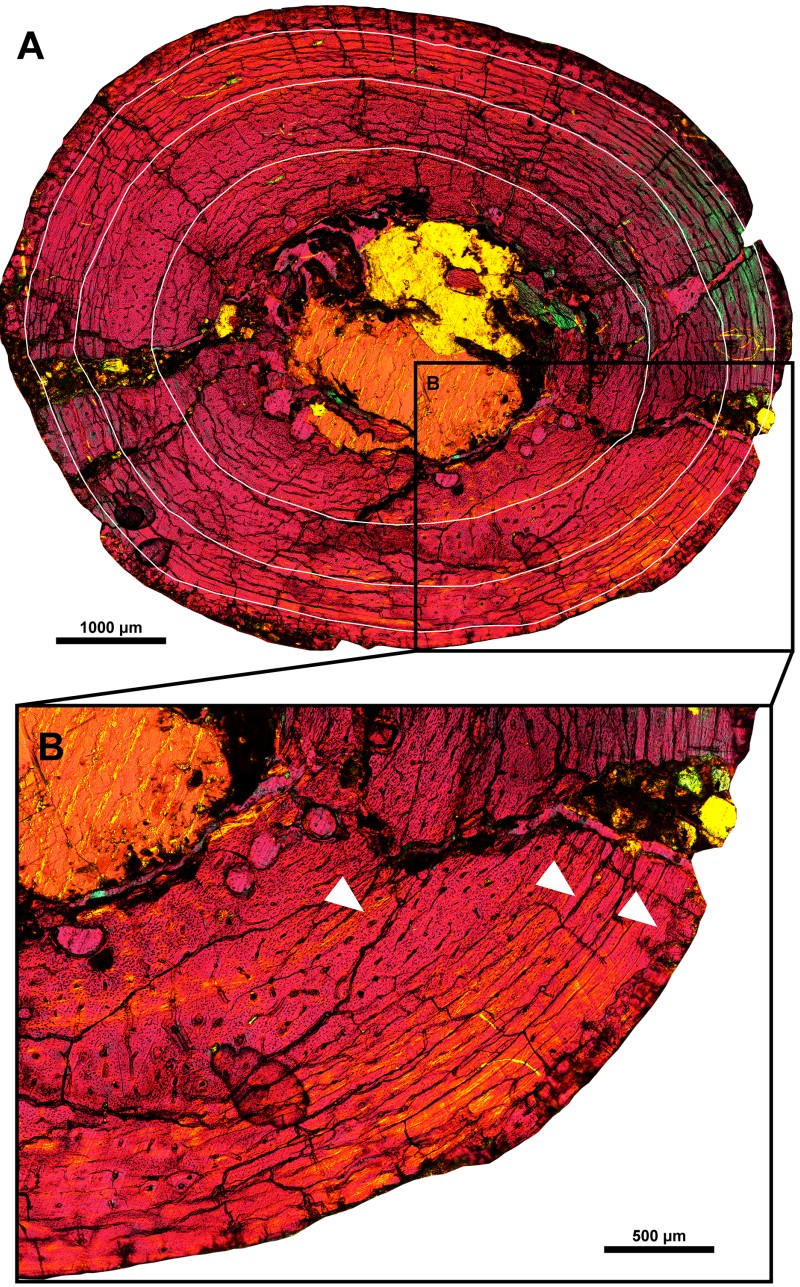

**Figure 9 Histological section of the right humeral shaft of *Yamaceratops dorngobiensis* (MPC-D 100/553).** Three LAGs are marked with white lines (A) and white arrowheads (B).

*2013*). In some regions of the cortex, LAGs are obliterated due to taphonomic or diagenetic alterations. The cortex ranges from 2.98 to 3.64 mm in thickness and does not contain secondary osteons or other signs of active bone remodeling. Unlike MPC-D 100/553, primary bone remodeling from individuals with three LAGs is reported from *Psittacosaurus lujiatunensis* and *Psittacosaurus sibiricus* (*Zhao et al., 2019*; *Skutschas et al.,*

*2021*), of which erosion bays, the first sign of primary bone remodeling, appeared from this stage.

## Body size and completeness

MPC-D 100/553 is the smallest and most complete *Yamaceratops* specimen to date and the first known articulated skeleton (*Makovicky & Norell, 2006*). Although the presence of additional *Yamaceratops* materials has been mentioned in the literature, they were not described (*Eberth et al., 2009*; *Nesbitt et al., 2011*).

MPC-D 100/553 is 67% the size of the holotype IGM 100/1315 based on the mandibular length (*Makovicky & Norell, 2006*: fig. 1) and is 62% the size of the referred specimen IGM 100/1303 based on postacetabular length of the ilium (*Makovicky & Norell, 2006*: fig. 18). MPC-D 100/553 is slightly smaller than '*Graciliceratops*' ZPAL MgD-I/156 based on the lengths of appendicular elements (*Maryańska & Osmólska, 1975*: table 2). The total body mass of MPC-D 100/553 was estimated based on the corrected equation for bipedal animals by *Campione et al. (2014)* and scaled following Developmental Mass Extrapolation (*Erickson & Tumanova, 2000*), recovering a body mass of 1.2 kg. The scaling was based on the assumption that the holotype is a fully-grown adult and that the minimum femoral circumference is isometric to skull length in *Yamaceratops*.

## Phylogenetic analysis

In the analysis of *Arbour & Evans (2019)*, unordered characters resulted in an extensive polytomy of most non-protoceratopsid basal neoceratopsians without a monophyletic Leptoceratopsidae. The new analysis with updated scorings (42 taxa and 257 characters) and unordered characters recovered both Leptoceratopsidae (except for *Helioceratops* and *Koreaceratops*) and Ceratopsoidea as a monophyletic clade (Fig. S4). However, the Protoceratopsidae had collapsed into a polytomy with some other basal neoceratopsians.

Removing one unstable taxon, *Helioceratops*, represented by fragmentary materials, resolved this polytomy, yielding a resolved Coronosauria (Protoceratopsidae and Ceratopsoidea) that is a sister clade to Leptoceratopsidae (except *Koreaceratops*), even in the unordered setting (Fig. S5). The phylogenetic analysis of 41 taxa and 257 characters produced a strict consensus tree (1,160 most parsimonious trees, tree length (TL) = 689 steps) with a topology similar to that of *Morschhauser et al. (2018c)* but with all characters unordered and no additional steps taken except for removing one unstable taxon (*Helioceratops*). It means that the extensive polytomy from the unordered matrix of *Morschhauser et al. (2018c)* and *Arbour & Evans (2019)* has been much resolved. The new analysis for the phylogenetic relationships of *Yamaceratops* among basal neoceratopsians recovered *Yamaceratops* as a sister taxon to the Leptoceratopsidae and Coronosauria combined (*i.e.*, Euceratopsia; *Madzia et al., 2021*) as suggested by *Morschhauser et al. (2018c)*. The main difference between the two strict consensus trees is the unresolved positions at the base of the tree and the positions of *Asiaceratops*, *Mosaiceratops*, and *Koreaceratops*. The unresolved position between *Hypsilophodon*, *Stegoceras*, and Ceratopsia, and between the Psittacosauridae, Chaoyangsauridae, and Neoceratopsia are expected to some extent since the relevant characters were not the main focus of the matrix

by *Morschhauser et al. (2018c)*, and the phylogeny of Cerapoda has not reached a consensus yet (*Han et al., 2018*; *Dieudonné et al., 2020*). In a cladogram of *Morschhauser et al. (2018c)*, *Asiaceratops* and *Mosaiceratops* formed a clade with *Yamaceratops*, sister to the Leptoceratopsidae and Coronosauria combined. In the new analysis, however, only *Yamaceratops* retained this position, and *Asiaceratops* and *Mosaiceratops* are in a position that is more derived than *Liaoceratops* and more basal than *Archaeoceratops*. In addition, in *Morschhauser et al. (2018c)*, *Aquilops* formed a clade with *Auroraceratops* and 'Graciliceratops' between *Archaeoceratops* and the clade, including *Yamaceratops*. But in our analysis, the clade has collapsed with the inclusion of *Koreaceratops*.

Removing two taxa, *Helioceratops* and *Koreaceratops* (scored only for postcranial characters), improved resolution and recovered the previously established monophyletic groups of Leptoceratopsidae, Coronosauria, Protoceratopsidae, and Ceratopsoidea (Fig. 10). The phylogenetic analysis of 40 taxa and 257 characters produced a strict consensus tree (730 most parsimonious trees, tree length (TL) = 688 steps) with a topology again similar to that of *Morschhauser et al. (2018c)* but with polytomies among basal neoceratopsians resolved. In this iteration, *Asiaceratops*, Mosaiceratops, *Auroraceratops*, and 'Graciliceratops' formed a clade in their position, and *Aquilops* was recovered as a more derived taxon than the clade of *Auroraceratops* and 'Graciliceratops' and more basal than *Yamaceratops*.

For the confirmation of the phylogenetic position of *Yamaceratops*, a second analysis was conducted using the character matrix of *Knapp et al. (2018)* as iterated by *Yu et al. (2020)*, with only scorings for *Yamaceratops* revised (Supplemental Information; Fig. S6). It also recovered *Yamaceratops* as the sister taxon to Euceratopsia (the clade formed exclusively by Leptoceratopsidae and Coronosauria; *Madzia et al., 2021*).

## DISCUSSION

### Taphonomy

The middle Javkhlant Formation was interpreted as representing a proximal alluvial plain environment in a seasonally-arid setting, of which coarse-grained deposits were interpreted as having deposited from alluvial channels, splays, and perhaps sheet floods (*Eberth et al., 2009*).

Taphonomic features of MPC-D 100/553 are as follows: (1) stage 0 degree of abrasion with unabraded bones (*Fiorillo, 1988*; *Cook, 1995*); (2) stage 1 weathering with bone-cracking from desiccation most apparent on the left lower jaw (*Behrensmeyer, 1978*; *Fiorillo, 1988*); (3) absence of root traces, borings, and gnaw/bite marks; (4) transverse/compression fracture breakage pattern (*Haynes, 1983*) that is most apparent on the right ilium, femur, and tibia; (5) plastic deformation and crushing present; (6) gleying around the skeleton present (*Jackson et al., 2018*).

The skeletal articulation, crouching posture, and the size of grains in the surrounding matrix (Figs. 2 and 11) indicate rapid perimortem to postmortem burial of MPC-D 100/553 (*Rogers & Kidwell, 2007*) with no transport, and that the carcass did not even float on water (*Syme & Salisbury, 2014*). A low degree of weathering of the bones and teeth

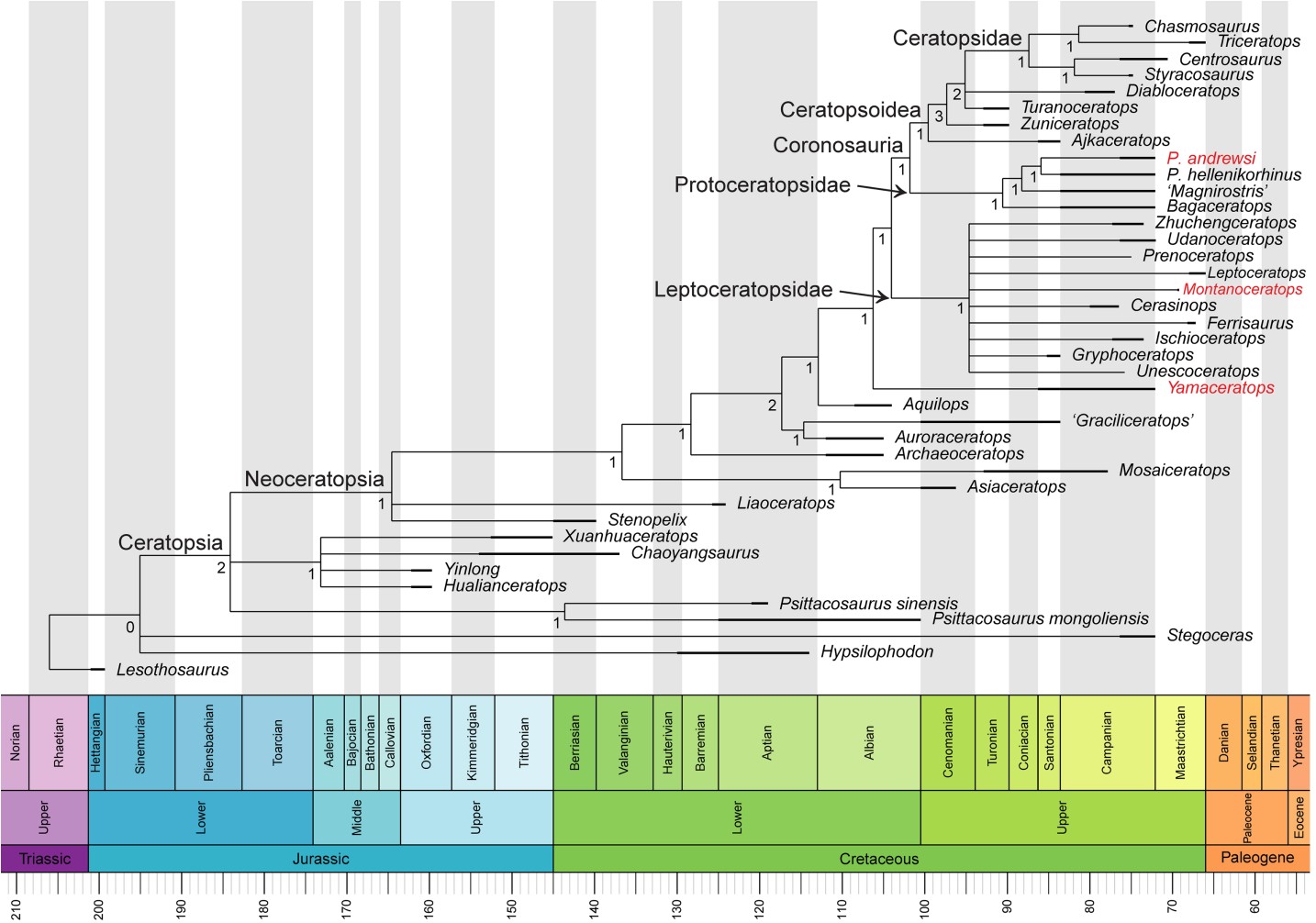

**Figure 10 Phylogenetic relationships of *Yamaceratops dorngobiensis* among ceratopsians using the *Arbour & Evans (2019)* matrix.** Strict consensus tree constructed by using the character matrix of *Arbour & Evans (2019)* (iteration of the *Morschhauser et al. (2018c)* matrix) with updated scorings for *Yamaceratops dorngobiensis* and ontogenetically variable characters unscored in taxa represented solely by juvenile specimens. Taxa with middle caudal neural spines about or more than four times longer than the centrum height are highlighted as red. Numbers at each node indicate Bremer support values. Time-calibrated using the R package "strap."

preserved in jaws supports limited surface exposure times (*Hill & Behrensmeyer, 1980*). The bone surfaces show no direct evidence of scavenging.

It is most likely that MPC-D 100/553 was preserved in this state by rapid burial following desiccation upon death. This interpretation is from a series of observations (Fig. 11). MPC-D 100/553 is notable in that: (1) the skeleton lies above a trough cross-bedding of fine- to coarse-grained sandstone with pebbles. (2) The dorsal side of the skeleton is facing the top of the strata. (3) The skeleton is articulated in a life-like crouching or sprawling position with its elbow pointing up. (4) The upper body is curved to the right, and the tail is strongly curved to the right, perpendicular to the pelvis and anterior portion of the tail. (5) The sclerotic ring and the supraorbital have collapsed but are still inside the orbit. (6) An isolated thin and narrow element, likely the posterior portion of the parietal bar, is preserved anteroventral to the distal fibula. (7) The right squamosal is

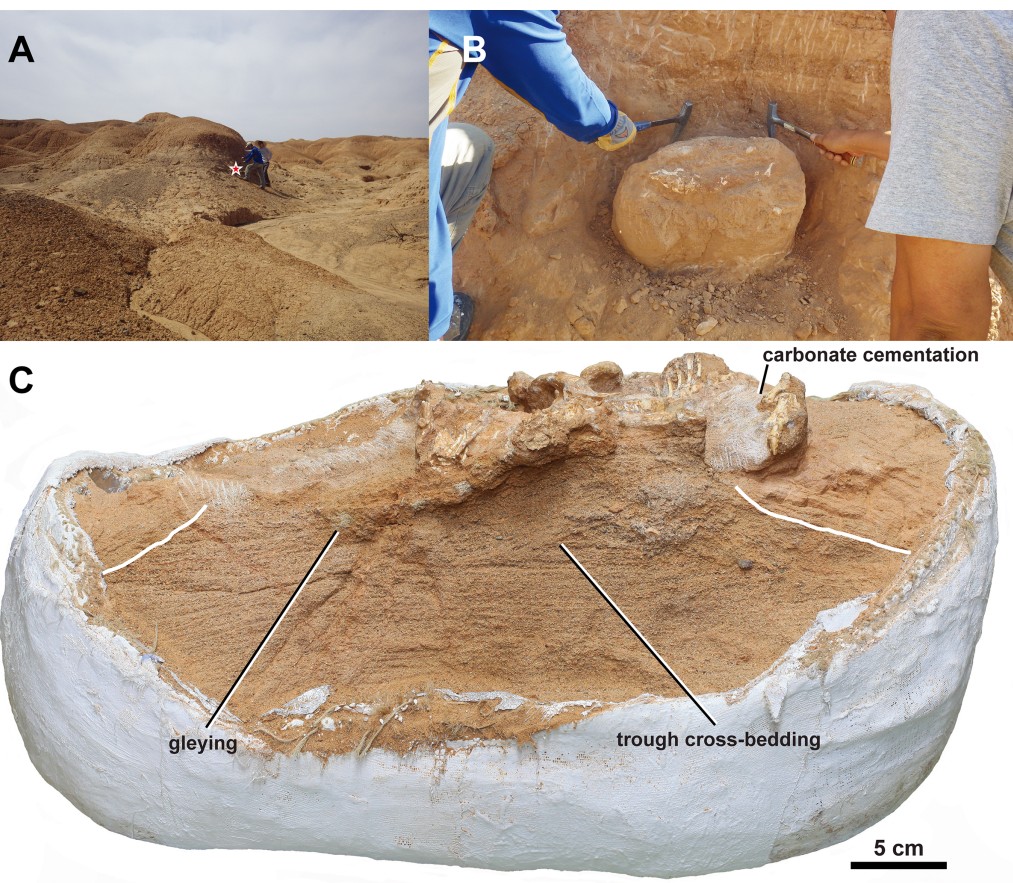

**Figure 11 Taphonomy of MPC-D 100/553.** (A) Photograph of the outcrop at the fossil locality. The red star indicates the position where MPC-D 100/553 was found; (B) photograph of MPC-D 100/553 exposed at the flank of a hill as found during the excavation; (C) MPC-D 100/553 (skull removed) and its underlying matrix with bedding preserved. The white lines mark the boundary between coarse- and fine-grained sandstone.

medially compressed with the ribs and probable pectoral elements placed where the left portion of the frill is expected to be, and the cervical series is missing. (8) The ribcage is collapsed, with the right ribs lying above the left ribs, with the ribs above the humeral head. (9) Gleying and carbonate cementation are present around the skeleton. (10) The skeleton is dorsoventrally crushed so that the maxillary teeth cover the dentary teeth, the jugal covers the surangular lateral ridge, the dorsal process of the jugal broken and displaced medially, the postorbital is facing dorsally rather than laterally, legs sprawling with transverse fractures on the femur, ilium disarticulated with its lateral surface facing up, ischium overturned with its medial surface facing up, and the right side of the tail is facing up. (11) Some conjugate tectonic joints cut across the skeleton and the surrounding matrix.

These indicate a taphonomic scenario of death at the lateral accretionary margin of a channel (1, 2, 3), followed by a short period of decay and desiccation of the body leading to breakage of the frill (4, 5, 6, 7, 8), subsequent rapid burial and post-burial microbial activities decomposing the skeleton (9) (*Allison, 1988*), compression of the skeleton during

sediment compaction and diagenesis (10), and final tectonic processes (11). These processes conform to the semi-arid seasonal environment of the middle Javkhlant Formation (*Eberth et al., 2009*).

MPC-D 100/553 is unlikely to have been preserved inside a burrow because the underlying fine- to coarse-grained sandstone with pebbles and trough cross-bedding (Fig. 11) are indicative of a fluvial deposit that had not been disturbed from burrowing. However, the laterally curved tail of MPC-D 100/553 may have resulted from drying the carcass during mummification, as has been suggested to have preserved *Auroraceratops* skeletons inside burrows (*Suarez et al., 2018*). Although MPC-D 100/553 shows the articulation of the skeleton in a life-like crouched posture with flexed hind limbs as in many skeletons preserved from miring layers (*Sander, 1992*; *Hungerbühler, 1998*; *Varricchio et al., 2008*; *Eberth, Xing & Clark, 2010*), the possibility of miring is also unlikely because of the underlying stratified sandstone without extensive mudstone.

The death pose and preservation of MPC-D 100/553 showing dorsoventral compression of the body with the skull and pelvis close to horizontal with legs folded and the tail turned laterally is likely due to the basal ceratopsian body plan. The laterally broad skull of mature basal ceratopsians may explain why most articulated skeletons have skulls that are not laterally compressed (with rare exemptions such as *Leptoceratops* CMN 8887 and *Psittacosaurus gobiensis* LH PV2; *Sternberg, 1951*; *Sereno, Xijin & Lin, 2010*). Similar posture can be seen in articulated skeletons of *Auroraceratops* (GSGM GJ <07>9-38; *Morschhauser et al., 2018b*), *Bagaceratops* (MPC-D 100/535; *Saneyoshi et al., 2011*), *Koreaceratops* (KIGAM VP 200801; *Lee, Ryan & Kobayashi, 2011*), *Leptoceratops* (CMN 8889; *Sternberg, 1951*), *P. andrewsi* (*e.g.*, AMNH 7417; *Gregory & Mook, 1925*), *Psittacosaurus* (*e.g.*, SMF R 4970; *Mayr et al., 2002*), and *Yinlong* (IVPP V14530; *Han et al., 2018*). *Psittacosaurus sinensis* (IVPP V738) was even preserved crouching with the tail curved (*Young, 1958*). But in this case, the tail is on top of the leg, unlike the MPC-D 100/553, where the right leg is stretched over the curved tail. Such specimens offer insights into post-mortem effects in perfectly articulated specimens.

The state of preservation in MPC-D 100/553 is reminiscent of an articulated *P. andrewsi* skeleton (MPC-D 100/512) from the "Fighting Dinosaur" specimen. The *P. andrewsi* skeleton MPC-D 100/512 is articulated in a crouching position. Its posterior dorsal vertebral column and tail are strongly curved, and the humeral head is placed beneath the ribs (*Barsbold, 1974*, *2016*) as in MPC-D 100/553. Another example of an articulated skeleton with a humeral head placed beneath the dorsal ribs is a partial *Bagaceratops* KID 196 (*Kim, Yun & Lee, 2019*), although in this case, the scapula and coracoid are articulated with the humerus, all beneath the dorsal ribs. The taphonomic process of MPC-D 100/553, MPC-D 100/512, and KID 196 having the ribcage on top of the proximal humerus can mainly be explained through the ventral collapse of the ribcage following decomposition. At the same time, the shoulder girdle and the forelimb remained relatively in position, resulting in the axial skeleton below the level of articulation with the appendicular skeleton (*Carpenter, 1998*; *Hone et al., 2014*). In MPC-D 100/512, from the maxillary teeth preserved below the alveoli and femur disarticulated from the acetabulum, it can be inferred that decay had taken place to a considerable degree (*Behrensmeyer &*

*Boaz, 1980*) and that the displacement happened after burial, to minimize displacement of disarticulated elements. It is worth noting that the holotype skull of *Yamaceratops* (IGM 100/1315) also preserves a maxillary tooth outside the alveolus, implying similar taphonomic processes.

A feature listed by *Sereno (2010*: 35) as an autapomorphy of *Psittacosaurus sinensis* is shared with MPC-D 100/553: "short lower jaw that positions the anterior margin of the predentary in opposition to the premaxilla rather than the rostral". However, the lower jaw in MPC-D 100/553 is more posteriorly positioned relative to the skull than in another *Yamaceratops* specimen IGM 100/1315, and this is likely due to deformation by the dorsoventral compaction of the skull in MPC-D 100/553, while IGM 100/1315 is less deformed (*Makovicky & Norell, 2006*). There seems to be variation among *Psittacosaurus sinensis* specimens (*e.g.*, IVPP V738 and BNHM BPV149, *Sereno, 2010*). Moreover, such a posteriorly positioned predentary in life is unlikely to have been functionally advantageous for *Psittacosaurus sinensis*, especially given that the premaxillae of *Psittacosaurus* were edentulous (*Sereno, 2010*), while in *Yamaceratops*, the premaxillary teeth occluded with the rhamphotheca of the predentary (Figs. 4 and 5; *Makovicky & Norell, 2006*). Therefore, the predentary of *Psittacosaurus sinensis* specimens positioned posterior to the general condition in ceratopsians may also be due to their taphonomy rather than a genuine taxonomic signal.

## Phylogenetic relationships of *Yamaceratops* with other basal neoceratopsians

A new comprehensive phylogenetic analysis including scorings for *Yamaceratops* from the new MPC-D 100/553 recovered *Yamaceratops* as a sister taxon to the Euceratopsia (Leptoceratopsidae plus Coronosauria; *Madzia et al., 2021*).

From our phylogenetic hypotheses, the presence of a large fenestrated frill could have been ancestral for the Leptoceratopsidae and Coronosauria and later lost in leptoceratopsid evolution.

*Yamaceratops* is the basalmost taxon with the maximum height of the caudal neural spine about or more than four times the height of the associated centrum among basal neoceratopsians, as the phylogenetic position of *Koreaceratops* likely falls within the Leptoceratopsidae (*Morschhauser et al., 2018c*). It indicates that the tall "leaf-shaped" (keeled) tail was perhaps plesiomorphic in leptoceratopsids and protoceratopsids and later lost in ceratopsids.

From the patterns of ontogenetic variation that are concordant in *Yamaceratops* and *P. andrewsi*, similar patterns could be expected for other basal neoceratopsians, and juvenile specimens could be more reliably incorporated into phylogenetic analyses (*Bhullar, 2012*).

Bipedalism of *Yamaceratops* (MPC-D 100/553) was predicted based on overall body proportions with long hindlimbs and tested from the quantitative method of *Chapelle et al. (2020)* using humeral and femoral circumferences. The measured minimum circumference of the humeral shaft is 23 mm, and that of the femoral shaft is 34 mm. Adding the data to the *Chapelle et al. (2020)* graph (fig. 1c) recovered a bipedal to equivocal position

close to a juvenile *Psittacosaurus lujiatunensis*–a taxon that went from quadrupedal to bipedal during ontogeny. The humeral shaft of MPC-D 100/553 is well preserved, but as the femur had been laterally crushed due to sediment compaction, its circumference is thought to have been slightly shortened. Therefore, assuming the actual minimum femoral circumference was longer than measured, *Yamaceratops* (MPC-D 100/553) fits into the range of bipedal animals. In a recent review, *Słowiak, Tereshchenko & Fostowicz-Frelik (2019)* concluded that many skeletal features of *Yinlong*, *Psittacosaurus*, *Liaoceratops*, *Mosaiceratops*, *Archaeoceratops*, and '*Graciliceratops*' were indicative of bipedality, while *Auroraceratops*, *P. andrewsi*, and leptoceratopsids were likely mainly quadrupedal, with a possible ontogenetic shift from facultative bipedalism to quadrupedalism in *P. andrewsi*. Our phylogenetic tree and bipedalism in *Yamaceratops* are mainly congruent with this conclusion.

## Ontogenetically variable features in *Yamaceratops*

The number of LAGs of the humerus suggests the relative age of this individual is at least 3 years old. The humerus contains mainly longitudinal and a small number of reticular and plexiform vessels. Blood vessel organization with a majority longitudinally directed suggests that bone mainly grew along its long axis. However, some degrees of radial, plexiform, laminar blood vessels exist in the humerus transverse section, suggesting circumferential growth rather than longitudinal in its final years. The tissue mainly consists of fibro-lamellar bone, as observed in modern vertebrates (*Horner & Padian, 2004*). Based on the absence of any secondary osteons and External Fundamental System (*Castanet, Newman & Saint Girons, 1988*; *Ponton et al., 2004*; *Klein, Scheyer & Tütken, 2009*; *Woodward, Horner & Farlow, 2011*), this individual was not somatically mature and was still growing.

Although a hierarchical analysis of the ontogeny in *Yamaceratops* is limited at this stage due to the small sample size of two skulls, the new skull offers a glimpse into the indicators of morphological maturity in basal neoceratopsians, as well as features diagnostic for this taxon.

MPC-D 100/553 is about 67% the size of the holotype specimen of *Yamaceratops* (IGM 100/1315) based on the mandibular length. Ontogenetically variable features in MPC-D 100/553 concordant with juvenile archosaurs that can be used in Marginocephalia (*Griffin et al., 2021*) include a relatively large orbit, long-grained surface texture on the femur, tibia, fibula, humerus, ulna, and radius (*Tumarkin-Deratzian, 2009*), open neurocentral sutures in every caudal vertebra (*Brochu, 1996*; *Hone et al., 2014*).

Juvenile features of MPC-D 100/553 shared with juvenile specimens of *Yinlong* (*Han et al., 2016*), *Liaoceratops* (*Xu et al., 2002*), *Auroraceratops* (*You et al., 2012*; *Morschhauser et al., 2018a*), *P. andrewsi* (*Brown & Schlaikjer, 1940*; *Handa, Watabe & Tsogtbaatar, 2012*; *Hone et al., 2014*), and *Bagaceratops* (*Maryańska & Osmólska, 1975*; *Czepiński, 2019*) include the relatively low angle of the lacrimal ventral ramus relative to the maxillary tooth row, a nasal that is flat and broad relative to the frontal, a relatively small jugal that does not flare laterally with its posterior ramus pointed posteroventrally, postorbital lateral

surface smooth, short predentary relative to the dentary length, dentary with a straight ventral edge and flat lateral surface ventral to the buccal emargination, the lateral ridge on the surangular low, and posterior parietal thin and narrow.

The holotype skull of *Yamaceratops* (IGM 100/1315) was described as being from a mature individual, from "closure of the sutures among the occipital and basicranial braincase elements" (*Makovicky & Norell, 2006*: 3). However, they also noted a possible indicator of immaturity: "the epijugal is not fused to the jugal, a feature that appears to be related to advanced maturity in *Protoceratops andrewsi* and ceratopsids" (*Makovicky & Norell, 2006*: 3). Compared to MPC-D 100/553, IGM 100/1315 certainly shows features of a more advanced ontogenetic stage: the high angle of the lacrimal ventral ramus relative to the maxillary tooth row; a relatively large jugal that flares laterally, with its posterior ramus pointed posteriorly; postorbital lateral surface rugose; dentary with a ventral edge that is convex in lateral view and curved laterally below the buccal emargination in anterior view; the lateral ridge on the surangular pronounced.

Additional variable feature in *Yamaceratops* that is related to ontogeny includes the shape and extent of the epijugal: the epijugal in IGM 100/1315 is crescentic and covers the posterior edge of the jugal to the ventral side (*Makovicky & Norell, 2006*), whereas the epijugal of MPC-D 100/553 is less curved and only covers the posterodorsal edge of the jugal and does not reach the ventral margin of the jugal. We hypothesize a similar pattern of jugal horn expansion during growth in other ceratopsians with epijugals, although specimens with articulated epijugals are relatively rare (*Horner & Goodwin, 2008*; *Morschhauser et al., 2018c*).

Many distinctive features in basal neoceratopsian taxa were acquired early in their developmental history. For example, in the very immature skull of *Bagaceratops* (ZPAL MgD-I/123) with a basal skull length of 37 mm, the following distinguishing features of this taxon are present: the presence of the accessory antorbital fenestra; the edentulous premaxilla; the fused nasals with a distinct bump; and the V-shaped buccal crest (*Maryańska & Osmólska, 1975*; *Czepiński, 2019*). From assessing the maturity of basal neoceratopsian specimens based on comparisons of the two *Yamaceratops* specimens, we decided that the character scorings for *Asiaceratops*, *Aquilops*, and 'Graciliceratops' had to be revised, following their probable immature status. However, no changes could be made for *Asiaceratops* because no ontogenetically relevant characters had been scored.

Although ontogenetic variation in ceratopsians has been described in *Psittacosaurus lujiatunensis*, *Liaoceratops*, *Montanoceratops*, *Protoceratops*, and *Bagaceratops*, they are either far away from the intermediate phylogenetic position of *Yamaceratops* or too fragmentary. Although multiple specimens from various ontogenetic stages have been recovered for *Yinlong*, *Psittacosaurus mongoliensis*, and *Auroraceratops*, their ontogeny has not been studied in detail. The discovery of an additional specimen of *Yamaceratops* from an even earlier or much later ontogenetic stage will provide information on the postnatal skeletal development of *Yamaceratops* and add insight into the general growth patterns in basal neoceratopsian dinosaurs.

## CONCLUSIONS

A new specimen of *Yamaceratops dorngobiensis* (MPC-D 100/553) was collected at the Khugenetjavkhlant locality from the Upper Cretaceous Javkhlant Formation. MPC-D 100/553 provides essential information on the anatomy of *Yamaceratops*, including autapomorphies which were unknown from the holotype (IGM 100/1315) and referred (IGM 100/1303; IGM 100/1867) specimens of *Yamaceratops*.

The new specimen is 67% the size of the holotype based on mandibular length. The differences between the holotype and the new specimen, likely representing ontogenetic variation in *Yamaceratops*, were concordant with patterns of ontogenetic variation known in *Protoceratops* (*Brown & Schlaikjer, 1940*). Therefore, MPC-D 100/553 was confirmed as a juvenile. The ontogenetically variable features present in both *Protoceratops* and *Yamaceratops* may be shared with other basal neoceratopsians or at least the group that is of intermediate phylogenetic position between *Yamaceratops* and *Protoceratops*.

Histological analysis of the humerus of MPC-D 100/553 showed no primary bone remodeling, confirming the immature ontogenetic stage of the animal. The presence of three LAGs indicates an age at death of approximately 3 years.

The new information on *Yamaceratops* and ontogenetically variable features in basal neoceratopsians recovered from MPC-D 100/553 were used in the new phylogenetic analysis. The analysis recovered *Yamaceratops* as a sister taxon to Euceratopsia (the most exclusive clade containing Leptoceratopsidae and Coronosauria; *Madzia et al., 2021*), meaning that *Yamaceratops* is the basalmost taxon with much-elongated caudal neural spines. During the evolution of ceratopsian dinosaurs, a change occurred in tail morphology, where the neural spines of the middle caudal vertebrae were elongated in basal neoceratopsians, reaching a peak in leptoceratopsids, protoceratopsids, and shortened in ceratopsids.

## INSTITUTIONAL ABBREVIATIONS

**AMNH**     American Museum of Natural History, New York, USA
**BNHM**     Beijing Natural History Museum, Beijing, China
**CAGS-IG**  Chinese Academy of Geological Sciences-Institute of Geology, Beijing, China
**CMN**      Canadian Museum of Nature, Ottawa, Ontario, Canada
**IGM**      Institute of Paleontology, Mongolian Academy of Sciences, Ulaanbaatar, Mongolia
**IVPP**     Institute of Vertebrate Paleontology and Paleoanthropology, Beijing, China
**KIGAM VP** Korea Institute of Geoscience and Mineral Resources, Vertebrate Paleontology, Daejeon, Republic of Korea
**MOR**      Museum of the Rockies, Bozeman, Montana, USA
**MPC**      Mongolian Paleontological Center, Mongolian Academy of Sciences, Ulaanbaatar, Mongolia

| OMNH | Sam Noble Oklahoma Museum of Natural History, Norman, Oklahoma, USA |
|------|------|
| PIN | Paleontological Institute, Russian Academy of Sciences, Moscow, Russia |
| RBCM | Royal British Columbia Museum, Victoria, British Columbia, Canada |
| TCM | The Children's Museum of Indianapolis, Indianapolis, USA |
| USNM | National Museum of Natural History, Smithsonian Institution, Washington D.C., USA |
| ZPAL | Institute of Paleobiology, Polish Academy of Sciences, Warsaw, Poland |

# ACKNOWLEDGEMENTS

Thanks go to all members of Gobi Dinosaur Supporters in 2014. Special thanks to Idersaikhan Damdinsuren for preparing histological sections. Drs. Jordan Mallon, an anonymous reviewer, and David Hone (Academic Editor) greatly improved this manuscript.

## Funding

This research is supported by the National Research Foundation of Korea (Grant Number 2019R1A2B5B02070240) to the corresponding author. The funders had no role in study design, data collection and analysis, decision to publish, or preparation of the manuscript.

## Grant Disclosures

The following grant information was disclosed by the authors:
National Research Foundation of Korea: 2019R1A2B5B02070240.

## Competing Interests

The authors declare that they have no competing interests.

## Author Contributions

- Minyoung Son conceived and designed the experiments, performed the experiments, analyzed the data, prepared figures and/or tables, authored or reviewed drafts of the paper, and approved the final draft.
- Yuong-Nam Lee conceived and designed the experiments, authored or reviewed drafts of the paper, and approved the final draft.
- Badamkhatan Zorigt performed the experiments, analyzed the data, prepared figures and/or tables, authored or reviewed drafts of the paper, and approved the final draft.
- Yoshitsugu Kobayashi conceived and designed the experiments, authored or reviewed drafts of the paper, and approved the final draft.
- Jin-Young Park performed the experiments, prepared figures and/or tables, authored or reviewed drafts of the paper, and approved the final draft.
- Sungjin Lee performed the experiments, prepared figures and/or tables, authored or reviewed drafts of the paper, and approved the final draft.

- Su-Hwan Kim performed the experiments, prepared figures and/or tables, authored or reviewed drafts of the paper, and approved the final draft.
- Kang Young Lee conceived and designed the experiments, authored or reviewed drafts of the paper, and approved the final draft.

## Data Availability

The raw data is available in the Supplemental Files.

## Supplemental Information

Supplemental information for this article can be found online at http://dx.doi.org/10.7717/peerj.13176#supplemental-information.

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
