# Peer review of "A new juvenile Yamaceratops (Dinosauria, Ceratopsia) from the Javkhlant Formation (Upper Cretaceous) of Mongolia"

_PeerJ, doi:10.7717/peerj.13176_

## Round 0.1 · original submission · Minor Revisions

Both referees are broadly supportive of the paper and note only relatively minor issues. I agree with the general points that they raised and that the manuscript can be improved by following their suggestions and the marked up documents that they have completed.

Reviewer 1 ·

Basic reporting

Missing literature in the references:

Line 77: "Norell and Barta, 2016" is not cited in the references.
Line 81: "Tereshchenko, 2020" is not cited in the references.
Line 83: "Tsogtbaatar et al., 2019" is not cited in the references.
Line 84: "Watabe et al., 2010" is not cited in the references.
Line 102: "Hone et al., 2014" and "Griffin et al., 2020" are not cited in the references.
Line 200: "You et al. 2007" is not cited in the references.
Line 358: "Butler et al., 2008" is not cited in the references.
Line 392: "Lambert et al., 2001" is not cited in the references.
Line 821: "Dieudonné et al., 2020" is not cited in the references.

Experimental design

No comment.

Validity of the findings

No comment.

Annotated reviews are not available for download in order to protect the identity of reviewers who chose to remain anonymous.

·

Basic reporting

This is generally a well-written article, and the osteological description is adequately detailed. There are several places in the ms where I find the phrasing somewhat awkward, as indicated in the attached markup. The ms is well-referenced, although I note that some of the references cited in-text are not also given in the Literature Cited section at the end (indicated in the markup). Where key comparisons are drawn to other taxa, emphasizing the uniqueness of the new specimen, I think some additional comparative figures would be useful. I note in the markup that the correspondence between figures 2A and B is not very good, and feel that the latter should probably be re-done. The lines overprint the LAGs in Figure 9, and should instead be indicated by arrows. Figure 11 is not especially aesthetic (especially compared to Figure 10), and should be redone to improve readability, perhaps by collapsing the genera that comprise Ceratopsidae. A table of standard measurements for the skeleton should be provided to facilitate comparison with other specimens. I would also like to see some additional background information provided about the Javkhlant Formation, which would help frame the discussion concerning the taphonomy of the new juvenile (the taphonomy of the specimen should likewise be placed first in the Discussion).

Experimental design

The methods employed here (cladistics, histology, CT scanning) are standard at this point, and not entirely concerning. Although the authors go to the effort of coding their new specimen into the matrix of Knapp et al., by their own admission, this matrix focuses heavily on the in-group relationships of Ceratopsidae. Therefore, it isn't entirely clear why the authors chose to code their specimen into this matrix. I wonder if the article might not be better served by relegating the results of the Knapp et al. analysis to a supplementary file, and limiting the in-text focus to the analysis of the Arbour and Evans matrix. This would both shorten the article and improve the readability. The authors should indicate how they calculated their support values in their cladograms. The changes made to prior codings for Yamaceratops should be summarized in a supplementary table. The authors should use developmental mass extrapolation when estimating the body size of their juvenile specimen because the standard femur circumference-body mass regressions apply to adults only.

Validity of the findings

I have no major issues with the validity of the findings. Given the weak support values on the cladogram(s), the authors should probably couch their conclusions about the placement of Yamaceratops relative to other neoceratopsians in more considered language. Some of the authors' anatomical interpretations appear to be 'off' in places (e.g., the preacetabular process of the ilium is visibly much shorter than the postacetabular process; the histology of the femur is described as both lacking evidence for remodeling and having it). The morphological significance of the parietal should be downplayed, given the authors' early admission that the identification of the bone is only tentative.

---

## Round 0.2 · accepted · Accept

Thank you for making these changes and I think the manuscript is much improved from its already strong starting point.